# Acoustic phonon recycling for photocarrier generation in graphene-WS$_2$ heterostructures

Ke Wei[1], Yizhen Sui[1,3], Zhongjie Xu[1,3], Yan Kang[1], Jie You[2], Yuxiang Tang[1], Han Li[1], Yating Ma[1], Hao Ouyang [1], Xin Zheng[2], Xiangai Cheng[1] & Tian Jiang [1✉]

Electron-phonon scattering is the key process limiting the efficiency of modern nanoelectronic and optoelectronic devices, in which most of the incident energy is converted to lattice heat and finally dissipates into the environment. Here, we report an acoustic phonon recycling process in graphene-WS$_2$ heterostructures, which couples the heat generated in graphene back into the carrier distribution in WS$_2$. This recycling process is experimentally recorded by spectrally resolved transient absorption microscopy under a wide range of pumping energies from 1.77 to 0.48 eV and is also theoretically described using an interfacial thermal transport model. The acoustic phonon recycling process has a relatively slow characteristic time (>100 ps), which is beneficial for carrier extraction and distinct from the commonly found ultrafast hot carrier transfer (~1 ps) in graphene-WS$_2$ heterostructures. The combination of phonon recycling and carrier transfer makes graphene-based heterostructures highly attractive for broadband high-efficiency electronic and optoelectronic applications.

[1] College of Advanced Interdisciplinary Studies, National University of Defense Technology, 410073 Changsha, P.R. China. [2] National Innovation Institute of Defense Technology, Academy of Military Sciences PLA China, 100010 Beijing, P.R. China. [3] These authors contributed equally: Yizhen Sui, Zhongjie Xu. ✉email: tjiang@nudt.edu.cn

Thermal heating has become the most critical topic for modern nanodevices, in which massive amounts of phonons are released through electron–phonon interactions and electrical energy is finally dissipated in the form of heat. Thus, new materials or structures are highly desirable for blocking such electron–phonon interactions or directly recycling the generated heat, in which case the heat production is reduced and the device power conversion efficiency is simultaneously significantly enhanced[1]. Graphene (G), an atomically thin carbon layer with a gapless band structure[2], a flat absorption feature[3], a high thermal conductivity and a low heat capacity[4], is considered a highly promising material for relieving this heat production[5–11] in the fields of light conversion and detection[12–20]. In recent years, both theoretical[21,22] and experimental[23,24] works have shown that carrier–carrier scattering in graphene is efficient enough to prevail over the electron-optical-phonon coupling, leading to highly efficient multiple hot-carrier generation originating from the primary photoexcited electron-hole pair. Photon energy can thus remain in the hot electron system[25–27] for further extraction instead of being transferred to the crystal lattice. However, as shown in Fig. 1a, such an adiabatic process can only last on a timescale of ~30 fs[24,28], during which the carrier–carrier scattering establishes a quasi-equilibrium distribution with an effective electron temperature. The following relaxation of the built-up hot carriers is dominated by the energy-dissipative optical phonon emission (~0.2 ps)[29,30] and the subsequent optical-acoustic phonon coupling (~2 ps)[30,31]. Thus, to acquire a device with both high performance and low heat generation, the excited hot carriers must be collected by the electrode within 0.2 ps before their energy is transferred to the optical phonons, which is very challenging even for graphene-based devices with an ultrahigh carrier mobility[32].

From a new perspective, phonon recycling is promising for tackling the above issue without the need for ultrafast detection. Specifically, in a newly designed unique structure, the heat generated by electron–phonon scattering can be reabsorbed rather than released into the environment. Phonon recycling has a long history, which can be traced back to the discovery of the thermoelectric and thermionic effects. Modern exploration of phonon recycling mainly concentrates on the design of various new recycling architectures, including thermophotovoltaics[33], photon-enhanced thermionics[34], hot-phonon absorption barriers[35] and phonovoltaics[36,37]. Despite these efforts, very little experimental evidence has been found to support the concept of phonon recycling. In particular, graphene, an excellent phonon transport material, has never been reported to show a phonon recycling property thus far.

Here, by establishing a van der Waals heterostructure, we utilize doped monolayer $WS_2$ to collect the acoustic phonon (lattice) energy generated in graphene and further deliver this phonon energy to the electron distribution via thermal excitation (Fig. 1b). This acoustic phonon recycling process is clearly recorded by spectrally resolved transient absorption (TA) microscopy, with a linear fluence-dependent amplitude and a fluence-insensitive characteristic time of > 100 ps. All these features can be well reproduced by an interlayer thermal transport model.

## Results

**Ultrafast electronic excitation and slow phononic excitation.** G-$WS_2$ heterostructures are prepared by mechanically exfoliating each component from bulk crystals and then separately transferring them onto a Si/SiO$_2$ substrate (see "Methods"). To trace the photocarrier relaxation dynamics, spectrally resolved TA microscopy is employed, as shown in Fig. 1c (see Methods for more details). This technique consists of two femtosecond pulses (~100 fs, 800 nm) with different intensities, with the stronger one being sent to an optical parametric amplifier (OPA) to change the photon energy, which can vary from 470 to 2600 nm, serving as the pump pulse. The weaker pulse, the probe beam, is focused onto a sapphire crystal after a controlled delay to produce supercontinuum white light and achieve broadband detection. Then, these two beams are combined and focused onto a single spot on the sample. The measured results are

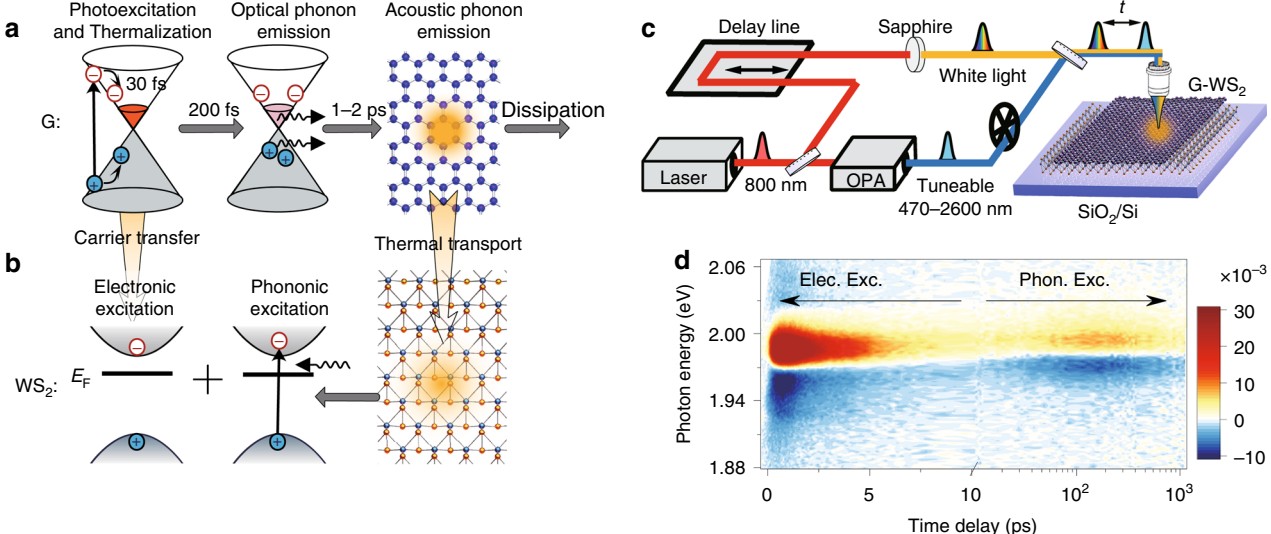

**Fig. 1 Experimental realization and results. a** Schematics of the electron and phonon dynamics in graphene after photoexcitation. The pump-excited electron-hole pairs thermalize in ~30 fs through carrier–carrier scattering, during which the absorbed photon energy remains in the electronic system. Subsequently, this energy is completely converted to lattice heat through optical (0.2 ps) and acoustic phonon (1–2 ps) emission. Without further treatment, the lattice heat will dissipate into the environment. **b** Monolayer $WS_2$ is utilized to collect the lattice heat in graphene through interfacial thermal transport and further deliver this lattice heat to the electronic system through thermal (phononic) excitation. In this heterostructure, a small portion of the initial hot carriers may directly transfer to $WS_2$ and induce electronic excitation. **c** Schematic of the TA measurement setup. **d** Typical 2D pseudo-colour TA map of the G-$WS_2$ heterostructure. Obvious electronic and phononic excitations of $WS_2$ are observed at timescales of <10 and >10 ps, respectively.

mathematically expressed in the form of the relative reflectance: $\Delta R/R_0 = (R - R_0)/R_0$, where $R$ and $R_0$ are the reflectance of the sample with and without pumping. In the linear absorption region, one can obtain $\Delta R/R_0 \propto \Delta n/n_s$[38], where $\Delta n$ is the difference in the carrier density with and without pumping, and $n_s$ is the saturation carrier density, with $\Delta n \ll n_s$. This linear relation between $\Delta R/R_0$ and $\Delta n$ allows us to trace the photocarrier relaxation dynamics by detecting the relative reflectance at different time delays.

Different carrier transfer dynamics can be obtained by simply exciting the G-WS$_2$ heterostructure with different pump photon energies. Specifically, if the photon energy is greater than the WS$_2$ optical bandgap (~2.02 eV, up-bandgap excitation), then photon absorption and photocarrier generation dominantly occur in WS$_2$, which has a much larger absorption coefficient than graphene. Naturally, the photocarriers transfer from WS$_2$ to graphene in the following relaxation process, leading to photoluminescence (PL) quenching of WS$_2$, with the quenching amplitude related to the interlayer coupling strength. In contrast, when the photon energy is less than the WS$_2$ optical bandgap (sub-bandgap excitation), WS$_2$ cannot be directly excited unless nonlinear absorption occurs, which requires a pump fluence level much higher than that used in our experiment. In the linear absorption region, although they cannot be directly pumped, the photocarriers in WS$_2$ still dramatically increase through various types of interfacial carrier transfer originating from graphene, including photothermionic emission[39], photothermoelectric[40] and bolometric[41] effects, interlayer charge-transfer (CT) transitions[42], and Förster-type energy transfer (see "Discussion" in Supplementary Note 2). These exciting processes are all mediated by the photocarriers belonging to the electronic excitation. In the G-WS$_2$ heterostructure, electronic excitation often occurs on a timescale smaller ~1 ps, during which the carriers in graphene are still hot. Subsequently, the transferred carriers in WS$_2$ completely decay within 10 ps due to the ultrafast nonradiative recombination, such as defect trapping or exciton-exciton annihilation, or simply back electron transfer to graphene. As a result, in the sub-bandgap pumping case, electronic excitation of WS$_2$ only contributes to the photocarrier response within 10 ps, which has been intensively studied in previous reports[42–46].

However, completely different phenomena are found in the TA measurement of our samples, as shown in Fig. 1d. Here, we present a 2D pseudo-colour TA map combining spectral (y-axis) and temporal (x-axis) resolutions. The data here and below are all obtained under excitation by a 0.48 eV, ~34 μJ cm$^{-2}$ pump pulse unless otherwise specified. In the spectral domain, the TA spectrum agrees well with the exciton resonance of WS$_2$, and it quickly decays to an undetectable level away from this energy. This indicates that the TA response is completely caused by the excited carriers/excitons in WS$_2$, while the signal from graphene is negligible in our experiments. The fact that excitations in WS$_2$, rather than graphene, account for the TA response is significant, as it confirms the existence of interlayer carrier transfer or/and phonon transport. The more interesting issue is revealed in the time domain, in which two TA relaxation features are found with completely different timescales. The initially fast feature, with its amplitude quickly peaking at ~0.5 ps and completely decaying within 10 ps, is attributed to the aforementioned electronic excitation. We have also provided a detailed discussion of this electronic excitation in Supplementary Note 2, where the possible approach of distinguishing the energy and charge transfers by carefully resolving the slight energy shift of the TA spectrum is presented. Below, we concentrate on the slow feature that occurs on the timescale of >10 ps. This slow feature, with an onset at ~10 ps and a maximum at ~150 ps, has never been found in

previous reports to the best of our knowledge, possibly owing to its relatively low amplitude and/or the nearly intrinsic nature of the general exfoliated WS$_2$ (discussed below). Nevertheless, this feature is significantly higher than the noise level. Additionally, it can be found in other samples (see below and also in Supplementary Fig. 14) and in all the sub-bandgap pumping cases (Supplementary Fig. 15). Regarding the origin of this robust slow feature, the above ultrafast electronic excitation can be safely ruled out due to the distinct timescale. We note that the formation of the interlayer exciton also induces a bleaching signal in the TA response[47]. This process is also very fast, with the induced carrier density peaking at ~1 ps and completely decaying at ~20 ps, completely different from the slow peaking time of ~150 ps in our measurement. As a result, we propose that this slow feature might be caused by the lattice heat generated from graphene, as the interlayer thermal transport is much slower than the carrier tunnelling.

Generally, phonons affect the TA signal through the following mechanisms: (i) Lattice heating-induced redshift of the exciton resonance[48]. Since the optical bandgap of WS$_2$ monotonically decreases with increasing temperature[49], lattice heating may cause a spectral shift of the exciton resonance and thus an antisymmetric derivative TA signal with a comparable photobleaching (PB) and photoinduced absorption (PIA) features, which seems very similar to our measurements (Supplementary Fig. 7). This lattice heating effect has been discussed in detail in Supplementary Note 4. To conclude, at least at current stage, we cannot completely prove or exclude this possibility. Further study on extracting the actual photocurrent of a G-WS$_2$ device is required to completely recognize this effect, as it cannot generate any photocarriers. Nevertheless, two indirect experimental results suggest that there should be other mechanisms contributing to this slow rising feature. First, within the numerous G-WS$_2$ heterostructures we prepared, not all of the samples show the same antisymmetric TA spectrum. Supplementary Fig. 8 presents a comparison of the TA spectrum from three different heterostructures, in which the third sample shows a completely asymmetric TA spectrum dominated by the PB feature, which can not be explained simply from the lattice heating effect. Second, the spectral response of another type of heterostructure, G-MoS$_2$, shows no secondly-rise TA peak at long delay time except for the initially ultrafast feature induced by electronic excitation (see Supplementary Note 8 for a detailed discussion). The absence of the slow TA feature cannot be explained by the crystal heating effect, since the ultrafast electronic excitation has already indicated a good interlayer contact, which in principle can support an efficient interlayer heat transport. (ii) The second mechanism is pump-induced acoustic wave or coherent lattice vibration. These effects induce an oscillating TA signal similar to in the case of a topological insulator[50,51]. Precisely, the signal oscillation originates from the direct excitation of the sample, which is not consistent with our result. The slow rising feature is only found in the sub-bandgap pumping case where only graphene can be excited and completely disappears when WS$_2$ is directly pumped using a 2.1 eV photon energy (Supplementary Fig. 3). Furthermore, when we couple another graphene layer on the other side of WS$_2$, forming a G-WS$_2$-G sandwich structure, the amplitude of the slow feature doubles (Supplementary Fig. 14). As a result, we propose that the slow rising TA peak is attributed to the phononic excitations from graphene, as depicted in Fig. 1a, b. Similar to the concept of photon recycling[52], we denote the phononic excitation process in question as acoustic phonon recycling (APR), considering that it delivers the acoustic phonon energy in graphene back into the electron distribution in WS$_2$.

**Theoretical model of the APR effect.** To gain more insights into the APR process, we proceed to a semiquantitative analysis of the relaxation dynamics of the APR-induced photocarriers. We begin with the exploration of the relation between the carrier density and the $WS_2$ lattice temperature. At the characteristic time of the phononic excitation (>100 ps), the electrons and lattice have reached thermal equilibrium in $WS_2$, and thus, the excited carriers approximately follow the Boltzmann distribution, $n = N_c \exp\left(-\frac{E_c - E_F}{k_B T_{L-WS_2}}\right)$, where $N_c$ is the effective electron density of states, $E_c$ and $E_F$ are the conduction band minimum energy and Fermi energy, $k_B$ is Boltzmann's constant and $T_{L-WS_2}$ is the $WS_2$ lattice temperature, with $E_c - E_F \gg k_B T_{L-WS_2}$. When the sample is excited by a femtosecond laser pulse, the lattice temperature and thus the carrier density vary with the time delay $t$. For a differential reflection configuration, the below relation is obtained:

$$\Delta R(t)/R_0 \propto -\Delta n(t)/n_s \cong \frac{N_c(E_c - E_F)\Delta T_{L-WS_2}(t)}{n_s k_B T_0^2}\exp\left(-\frac{E_c - E_F}{k_B T_0}\right) \tag{1}$$

where $T_0 = 300$ K is the ambient temperature and $\Delta T_{L-WS_2}(t) = T_{L-WS_2}(t) - T_0$ is the pump-induced temperature rise of $WS_2$ lattice, with $\Delta T_{L-WS_2} \ll T_0$. Here, we ignore the changes in $E_c$ and $E_F$ since they are weakly dependent on $T_{L-WS_2}$. Therefore, it is concluded that the relative reflectance is dominated by two factors, namely, the Fermi energy $E_F$ and the lattice temperature rise $\Delta T_{L-WS_2}$.

In fact, without special treatment, such as impurity or electrical gate doping, the general as-prepared $WS_2$ is slightly $n$ doped due to the sulfur vacancies or/and $SiO_2$/Si substrate doping. Despite this unintentional doping, $E_F$ is still located at an energy level near the intrinsic state, far from $E_c$, leading to a negligible phononic excitation. However, considering the exponential dependence of $\Delta R/R_0$ on $E_c - E_F$, a substantial increase of $E_F$ by doping $WS_2$ may induce a significant enhancement of the phononic excitation. Thus, it is necessary to characterize the doping level of $WS_2$ to understand the phononic excitation mechanism.

Photoluminescence spectroscopy is viewed as a powerful and simple technique to qualitatively determine the Fermi energy of transitional metal dichalcogenides (TMDCs)[53]. Specifically, if the sample is nearly intrinsic, with a very low free carrier concentration, then the PL emission will be dominated by the neutral exciton ($A^0$), as no extra free carrier can be combined with it. If the Fermi energy is raised away from the neutral level, then the exciton may be charged, becoming a charged trion ($A^-$). This charging process is reflected in the PL spectrum through a redshift of the emission peak, with a shift amplitude of ~26 meV (the trion binding energy). Moreover, since $A^-$ has a much larger nonradiative recombination rate than $A^{0,53}$, the conversion of $A^0$ to $A^-$ is accompanied by a drastic decrease in the PL intensity. After determining the influence of the Fermi energy on the PL spectrum, we carry out PL imaging experiments on the heterostructures, as shown in Fig. 2a (Sample 1). The sample mainly consists of three different regions: monolayer $WS_2$, G-$WS_2$ and G-$h$-BN-$WS_2$. The $h$-BN spacer (4–5 layers) in the last region serves two purposes: blocking the electronic excitation (electrical insulation) and sustaining the phononic excitation (thermal conduction), which will be discussed later. Here, the PL intensities are quenched ~200 and ~4 times in the G-$WS_2$ and G-$h$-BN-$WS_2$ heterostructures compared to that in monolayer $WS_2$ (Fig. 2b), along with different amplitudes of the PL peak redshifts (Fig. 2c). Despite this PL quenching and peak shift, the $A^0$ and $A^-$ components cannot be distinguished simply by fitting the PL

spectra, as the large spectral broadening and the band shift due to other factors (e.g., the change in the dielectric environment) may significantly affect the fitting accuracy. To address this issue, the TA spectrum is used to determine the resonant energies of $A^0$ and $A^-$, as it generally has a narrower linewidth than the PL emission. The up-bandgap pumping configuration is used (2.1 eV, 7.8 μJ cm$^{-2}$) to directly excite the $WS_2$ monolayer, as shown in Fig. 2d. The $A^0$ and $A^-$ components are clearly separated in the TA spectra for all the three regions, with a fitted trion binding energy of ~25 meV, well consistent with a previous report[54]. By comparing the PL and TA spectra, we can find that $A^0$ emission dominates in monolayer $WS_2$, indicating a nearly intrinsic state of this region, while the PL spectra in the other two heterostructures are both governed by $A^-$ emission, implying a tremendous increase in the Fermi energy relative to the intrinsic level ($E_{F0}$). This Fermi energy change is most likely caused by impurity or defect doping during sample preparation, or simply caused by interlayer electron transfer from graphene to $WS_2$. Nevertheless, this doping may significantly reduce the value of $E_c - E_F$ in Eq. (1) and enhance the photocarrier density excited by the APR process.

The other factor that determines the APR amplitude is the $WS_2$ lattice temperature increase ($\Delta T_{L-WS_2}$), which is time delay dependent after photon excitation. To simulate the relaxation of $\Delta T_{L-WS_2}$, we develop an interfacial thermal transport model based on the electron and phonon relaxation pathways (see discussions in Supplementary Note 3). Briefly, under femtosecond pulse pumping, the ultrafast excitation and thermalization of the hot carriers in graphene is regarded as an initial condition, providing an initial hot carrier temperature of $T_1$, which can be as high as thousands of Kelvins. $T_1$ can be quantitatively estimated by integrating the energy relaxation equation of a hot electron gas[35] $dE_{e-G}/dt = C_{e-G}(dT_{e-G}/dt) = P_{in}\delta(t)$, giving $T_1 = \sqrt{T_0^2 + 2P_{in}/\gamma}$, where $E_{e-G}$ is the hot electron energy, $C_{e-G} = \gamma T_{e-G}$ is the electronic heat capacity in graphene, $\gamma$ is the 2D Sommerfeld constant, and $P_{in}$ is the pump fluence absorbed and delivered to the electrons. After thermalization, the hot carriers cool down to the Fermi level via phonon emission, which is characterized by a rate equation of $C_{e-G}(dT_{e-G}/dt) = -H_{e-G}$, where $H_{e-G}$ represents the electron heat loss rate. Notably, the graphene carrier cooling process has been intensively investigated in previous reports, with different mechanisms for the temperature-dependent heat loss rate being proposed, including intrinsic acoustic phonons[55], interactions with remote surface polar phonon modes[56], and disorder-enhanced supercollisions with acoustic phonons[57,58]. Here, since we mainly concentrate on the thermal transport process occurring on a timescale >10 ps, the ultrafast carrier cooling within 1 ps does not affect the results. Thus, for simplicity, we use the supercollision model with $H_{e-G} = A(T_{e-G}^3 - T_0^3)$. The subsequent processes comprise graphene lattice heating, interfacial thermal transport, $WS_2$ lattice heating and heat transfer to the Si/$SiO_2$ substrate (Supplementary Fig. 6), which can be fully explained by thermal conductance-limited heat transport. Based on this model, the relaxation dynamics of the pump-induced $WS_2$ lattice temperature rise and thus that of the phonon-excited carriers can be directly estimated.

Figure 2e shows a comparison between the measured and simulated photocarrier relaxation dynamics. The excellent agreement between them confirms the thermal transport from graphene to $WS_2$. To profoundly analyse the APR process, the simulated relaxation dynamics of the electron and lattice temperatures in graphene and $WS_2$ are presented in Fig. 2f under the same excitation conditions as in Fig. 2a. Immediately after excitation, the graphene carrier temperature $T_{e-G}$ increases to thousands of Kelvins, followed by an ultrafast decay (~0.5 ps) due to phonon emission. Since the interfacial carrier transfer

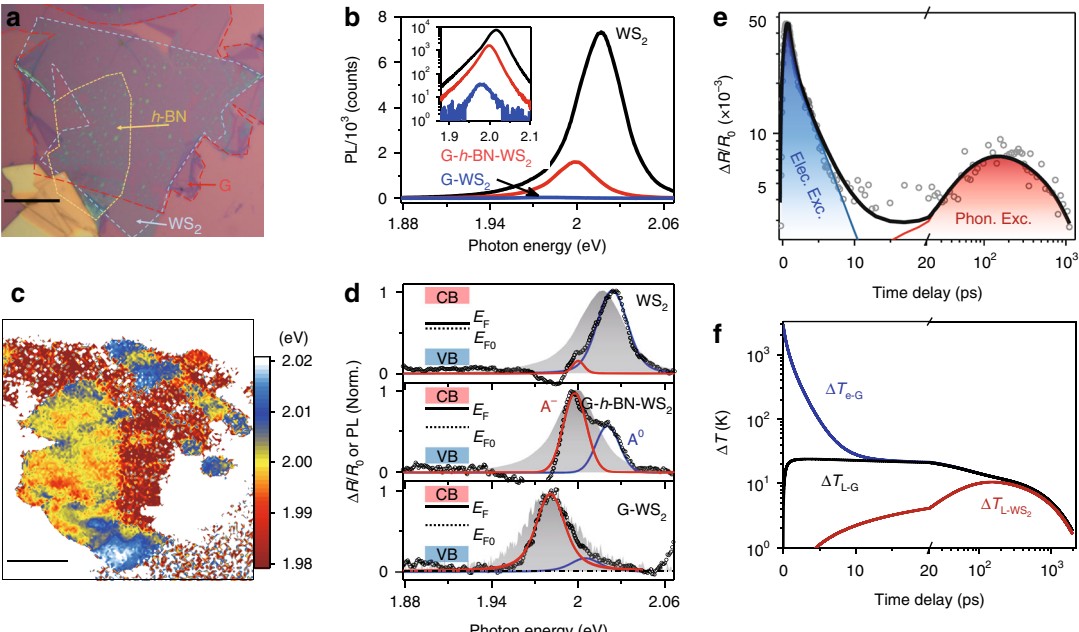

**Fig. 2 APR process induced by the rising Fermi energy. a** Optical image of the graphene-based heterostructure (Sample 1). Scale bar, 10 μm. **b** PL intensity for different sample regions of **a**. Inset: The same data with a log y-axis. **c** PL peak energy image of the heterostructure. Scale bar, 10 μm. **d** Comparison of the normalized PL (grey area) and TA (black dots) spectra. The TA spectra are pumped by an up-bandgap photon energy (2.1 eV) and fitted based on a neutral exciton ($A^0$, blue line) and a charged trion ($A^-$, red line), with an energy shift (trion binding energy) of ~25 meV. (Inset) Illustration of the Fermi level position in different sample regions. **e** Comparison of the measured (dots) and simulated (line) photocarrier dynamics. The blue and red areas represent the electronic and phononic excitation components, respectively. **f** Thermal transport model-predicted carrier cooling process in graphene ($T_{e-G}$), and lattice temperature relaxation in both graphene ($T_{L-G}$) and WS$_2$ ($T_{L-WS_2}$). For **e**, **f**, the pump energy is 0.48 eV, and the pump fluence is ~34 μJ cm$^{-2}$.

from graphene to WS$_2$ is highly dependent on $T_{e-G}$, the transfer process mainly occurs in the first 0.5 ps after excitation, resulting in an initial increase in the differential reflectance. Note that in the following time delays from 1 to 10 ps, interfacial carrier transfer from graphene to WS$_2$ is negligible, while the nonradiative recombination of the transferred carriers in WS$_2$ or/and back electronic transfer from WS$_2$ to graphene play the dominant role. Thus, the carrier decay rate on this timescale is dependent on the recombination or/and back transfer process rather than on $T_{e-G}$.

The graphene carriers and lattice are in complete thermal equilibrium within ~10 ps, leading to a maximum lattice temperature rise of $\Delta T_{L-G(max)} \cong 25$ K. After thermal equilibration, the subsequent processes consist of interfacial thermal transport from graphene to WS$_2$ and heat dissipation from WS$_2$ to the Si/SiO$_2$ substrate, which may induce a relatively slow increase (~35 ps) and then a decay (~1 ns) of the WS$_2$ lattice temperature, with a maximum increase of ~11 K peaking at ~150 ps. This slow relaxation dynamics is governed by the thermal conductance at the G/WS$_2$ ($\Gamma_{G-WS_2}$) and WS$_2$/SiO$_2$ ($\Gamma_0$) interfaces, which are fitted to be $5 \pm 2$ and $1.0 \pm 0.3$ MW m$^{-2}$ K$^{-1}$, respectively. The value of $\Gamma_{G-WS_2}$, although 3~4 times smaller than that at the $h$-BN-MoS$_2$ interface[59], is within the experimental error since the contact thermal conductance is highly sensitive to the coupling strength and the impurities/defects at the interface. The relatively low thermal conductance at the WS$_2$/SiO$_2$ interface may be attributed to the low out-of-plane thermal conductivity of WS$_2$[60], which can efficiently confine the thermal energy in WS$_2$ for carrier re-excitation, rather than it being dissipated into the substrate.

**Spacer tuning of the APR effect.** One of the hallmarks by which APR differs from interfacial carrier transfer is that the energy recycling relies on phonon transport rather than on carrier transfer. Thus, if we insert a spacer with both electrical insulating and thermal conducting properties, such as atomic layer $h$-BN[61], then the interfacial carrier transfer will be significantly suppressed, while the APR process may still be allowed. In contrast, when a spacer with electrical conducting and thermal insulating properties is used, such as WSe$_2$[62,63], the phonon transport will completely disappear. To confirm this assumption, 4–5 layer $h$-BN and monolayer WSe$_2$ are inserted between the graphene and WS$_2$ in Sample 1 (Fig. 2a) and Sample 2 (Supplementary Fig. 2), respectively, with the TA measurement results shown in Fig. 3a, c. To exclude the sample differences, the case without a spacer (G-WS$_2$) is also plotted for comparison. Obviously, the APR process is only observed in the G-$h$-BN-WS$_2$ and G-WS$_2$ regions and completely disappears in the G-WSe$_2$-WS$_2$ heterostructure, even though the thickness of WSe$_2$ (monolayer) is much smaller than that of $h$-BN (4–5 layers).

Figure 3b, d shows the relaxation dynamics of the photocarriers. For the heterostructure with the WSe$_2$ spacer, the APR process is negligible due to the poor interfacial thermal transport, while for the $h$-BN case, obvious phononic excitation is found with a peaking time of ~400 ps, much longer than that in the direct contact case (~150 ps in G-WS$_2$). Such slow relaxation dynamics is well described using the above APR model, with the two parameters of $\Gamma_{BN-WS_2} = 4 \pm 2$ MW m$^{-2}$K$^{-1}$ and $\Gamma_0 = 0.8 \pm 0.3$ MW m$^{-2}$K$^{-1}$, which are in good agreement with the values obtained from the heterostructure without spacers. Here, we note that the $h$-BN spacer only slightly reduces the $\Delta R/R$ amplitude by ~1.3 times, while the APR model predicts an ~5 times decrease in $\Delta T$ due to the relatively large thermal capacity of 4 layer $h$-BN. This discrepancy may be caused by the change in the WS$_2$ Fermi energy-related scaling factor in Eq. (1). Nevertheless, the good agreement of the carrier relaxation dynamics between the experimental results and thermal transport model in

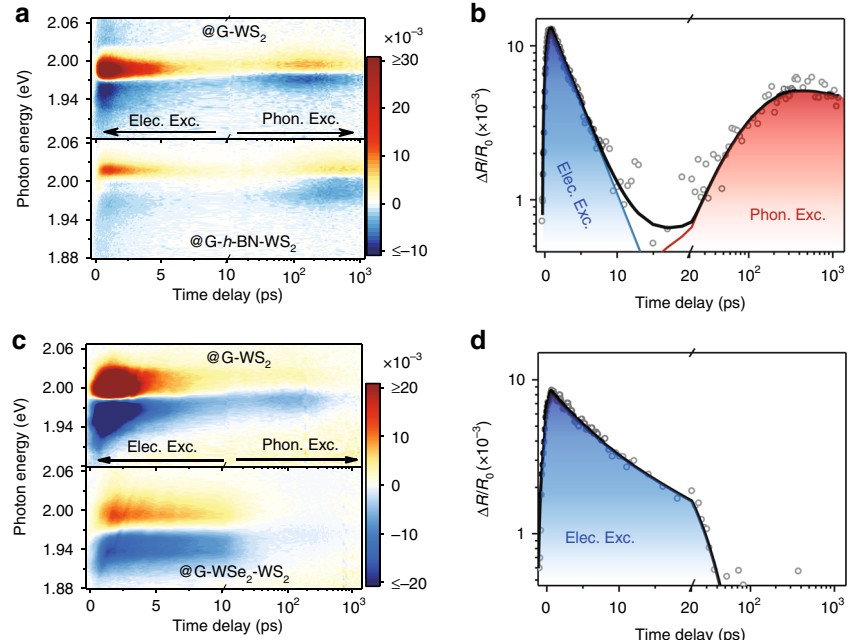

**Fig. 3 Tuning of the APR process by inserting spacers. a** 2D pseudo-colour TA maps of the G/WS$_2$-based heterostructure with (bottom, G-$h$-BN-WS$_2$) and without (top, G-WS$_2$) $h$-BN spacer. The data for G-WS$_2$ are extracted from Fig. 1d. (**b**) Photocarrier relaxation dynamics measured in the heterostructure with the $h$-BN spacer. Dots are experimental results extracted from **a**, while solid lines are the results calculated using the APR model, with the electronic (phononic) excitation component labelled by the blue (red) area. **c, d** Same data as in **a, b** but measured in heterostructures with and without WSe$_2$ spacer. The phononic excitation completely disappears in the heterostructure with a WSe$_2$ spacer due to its poor thermal transport property. The upper (lower) panel is measured from Sample 1 (2).

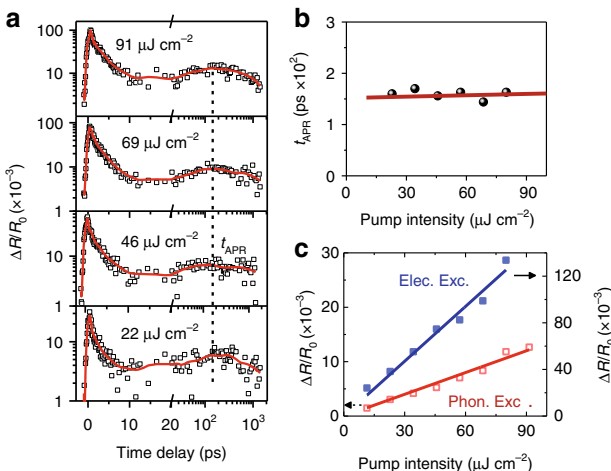

**Fig. 4 Pump fluence-dependent APR characteristics. a** Differential reflectance kinetics at different pump fluences. Dots are measurement results, and solid lines are smooth data for clarification. **b** Extracted APR characteristic time as a function of pump fluence. **c** Electronic (blue) and phononic (red) excitation intensities as a function of pump fluence. In **b, c**, dots are measurement results, and solid lines are fitting results from the APR model.

both the direct contact and $h$-BN spacer cases undoubtedly demonstrates the interlayer thermal transport.

**Intrinsic properties and efficiency of the APR effect.** Considering the practicality of the APR effect, we next show its properties under different excitation intensities. Figure 4a shows a typical example measured from the G-WS$_2$ heterostructure. Similar results for the heterostructure with the $h$-BN spacer are

also provided in Supplementary Fig. 16. Over a broad pump fluence range of 11–91 μJ cm$^{-2}$, the shape of the carrier relaxation curves remains unchanged. This robust feature indicates that the characteristic properties of the APR effect, such as the transfer rate and the lifetime, are insensitive to the pump fluence. To quantitatively analyse this phenomenon, we extract the peak delay time of APR-induced photocarriers ($t_{APR}$) at different incident pump fluences and compare them with the results calculated based on the above model, and the results are shown in Fig. 4b. The simulated results exhibit great agreement with the experimental data, which show that a 10 times pump fluence enhancement only induces an ~5% delay of $t_{APR}$. Such a robust feature may open up new paths for optoelectronic applications with high working power or a high dynamic range, such as photodetectors and bolometers.

Proceeding with the analysis, a linear-like relation between the amplitude of APR-induced photocarrier density and the pump fluence is found, as demonstrated in Fig. 4c. This can be reasonably explained by the thermal transport model. Since the APR characteristic time is greater than 100 ps, the whole system, including the graphene carriers and lattice, WS$_2$ lattice and Si/SiO$_2$ substrate, has reached thermal equilibrium. The fraction of the absorbed photon energy delivered to the WS$_2$ lattice is only determined by the heat capacity of each component and the thermal conductance at each interface, instead of the pump fluence. Thus, it can be safely concluded that in the linear absorption region ($\Delta T_{L-WS_2} \ll T_0$), the interfacial thermal transport efficiency in a given heterostructure is approximately independent of the pump intensity.

The APR efficiency is an alternative key factor for practical applications. However, accurate calculation of the internal quantum efficiency of the APR process is very difficult since it is exponentially correlated to the WS$_2$ Fermi energy according to Eq. (1), which may be easily influenced by both internal and

external factors, such as chalcogenide vacancies, impurities, the substrate or electrostatic doping. Thus, we only provide a comparative discussion on the efficiency of the APR process (>10 ps) and that of interfacial carrier transfer (< 10 ps). It is important to mention that the photocurrent response of a similar G-WSe$_2$-G structure reported by Massicotte[39] was dominantly attributed to the electronic excitations induced by the photo-thermionic effect, while the APR-induced phononic excitation was not discussed in detail. From Fig. 4c, one can find that although the peak amplitude of the slow rising feature is ~13 times smaller than that of the carrier transfer, the steady and prolonged heat flows may make considerable contributions to the electrical response of the device. If the crystal heating effect is negligible, the slow rising TA feature can be completely attributed to the APR process, which will play a more important role (i.e., ~50 times) than the ultrafast carrier transfer when considering the temporal integration of the reflectance signal. Therefore, further study on the actual photocurrent response under sub-bandgap photon excitation is required, which is not only of theoretical significance, but also useful for the optimization of practical device efficiency.

In addition to the pump fluence, other external factors that may affect the APR efficiency have also been systematically studied, e.g., the interfacial coupling strength (Supplementary Note 5), the twist angle between graphene and WS$_2$ (Supplementary Note 6), the number of WS$_2$ layers (Supplementary Note 7), and even the use of different types of TMDC materials such as MoS$_2$ (Supplementary Note 8). Detailed discussions can be found in the Supplementary materials, and we will not elaborate the results here.

Equation (1) and the thermal transport model further furnish a clue for improving the APR efficiency. Considering that the photocarrier density is exponentially dependent on the WS$_2$ Fermi energy, electrical gate control may be an efficient strategy to improve the APR efficiency. For instance, if $E_c - E_F$ decreases from 0.2 to 0.1 eV, then the relative carrier density may significantly increase from 0.001 to 0.027 under the same lattice temperature rise of $\Delta T_{L-WS_2} = 10$ K. Moreover, the thermal transport model suggests that the APR efficiency can also be greatly enlarged by increasing the thermal conductance at the G/WS$_2$ interface and simultaneously reducing that at the WS$_2$/substrate interface. In this way, the absorbed photon energy can be efficiently confined in WS$_2$ and further transferred to the photocarriers. Finally, by replacing WS$_2$ with another semiconductor material with a smaller lattice heat capacity and a narrower bandgap, the values of $\Delta T_{L-WS_2}$ and $E_c - E_F$ in Eq. (1) can also be efficiently increased and decreased, respectively, leading to a significant increase in the APR efficiency.

## Discussion
In summary, we have found an acoustic phonon recycling process in G-WS$_2$ heterostructures by using microscopic broadband transient absorption spectroscopy. During this process, the photogenerated heat in graphene is efficiently transferred to highly doped WS$_2$ and then re-excites the electron-hole pairs in WS$_2$ for further extraction. Benefiting from the gapless nature of graphene, this APR effect is observed over a very broadband pump photon energy below the WS$_2$ optical bandgap. The photocarriers excited by the APR process can be well predicted with an interfacial thermal transport model, with a sufficiently long characteristic time of >100 ps for carrier extraction. Furthermore, the photocarrier density is expected to be highly sensitive to the doping level of WS$_2$, enabling effective manipulation of the APR efficiency by simple electrostatic doping. The APR effect reported here can not only reduce general heat production but also significantly enhance the photocarrier generation in modern nanoelectronic and optoelectronic devices.

## Methods
**Heterostructure fabrication.** The heterostructures were fabricated using the dry transfer process, as described in ref. [64]. Monolayer graphene, WS$_2$, WSe$_2$ and MoS$_2$ were confirmed by using a combination of photoluminescence and Raman spectra, while the exact thickness of the $h$-BN spacer was measured by atomic force microscopy (Innova, Bruker).

**Microscopy broadband femtosecond pump-probe system.** Transient absorption measurements were performed using a homebuilt pump-probe system operating in reflective mode. The pump beam, with wavelengths ranging from 470 to 2600 nm, was obtained by pumping an optical parametric amplifier (TOPAS) with the output of a Ti:sapphire laser (Spectra-Physics, 800 nm, 1 kHz). A 1 kHz (amplifier stage) rather than an 80 MHz (oscillator stage) pulse frequency[65] was used here to avoid the potential heat accumulation effect induced by the previous pump pulses[50]. A small portion of the Ti:sapphire laser was directed into a sapphire crystal to generate supercontinuum white light ranging from 470 to 1100 nm, which worked as the probe beam. To improve the signal-to-noise ratio, before reaching the sample, the white light was equally divided into two beams by a 50/50 beamsplitter, forming the signal and reference beams. The reference beam was directly sent to a fibre-coupled multichannel spectrometer, while the signal beam was first overlapped with the pump beam by a 10/90 beamsplitter. Then, both were focused onto the sample with a ×50 long-focus objective, with a signal beam size of ~2 μm and a pump beam size of 2-50 μm, varying with the pump wavelength. The reflection of the signal beam backtracked to the 50/50 beamsplitter and was then also sent to the multichannel spectrometer, with a spectral resolution of ~0.1 nm. Finally, both the signal and reference beams were detected by a CCD camera. The dynamics of the photoinduced signal were realized with a computer-controlled delay line in the probe path. To ensure a good signal-to-noise ratio, the TA signal was acquired by averaging the data from 120 k spectra.

**PL spectrum measurements.** The PL spectrum was excited by a-532nm solid-state laser, collected by a confocal microscope (LEICA DM 2700 M), and recorded by a spectrometer (ANDOR SR-500i-B1-R) equipped with a CCD detector. The spatially resolved PL mapping was acquired by scanning the excitation beam with a 2D galvanometer, with a spatial resolution of ~0.3 μm.

## Data availability
The data that support the findings of this study are available from the corresponding author upon request.

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

## Acknowledgements

We are grateful for financial support from the National Natural Science Foundation of China (11804387, 11802339, 11805276, 11902358, 61805282, 61801498); the Scientific Researches Foundation of National University of Defense Technology (ZK18-03-22, ZK18-01-03, ZK18-03-36); the Science Fund for Distinguished Young Scholars of Hunan Province (2020JJ2036).

## Author contributions

T.J., K.W. and Z.X. conceived the idea and designed the research; Y.K., K.W. and Y.M. fabricated all the samples; K.W. and Y.S. performed the optical measurement. H.O. performed the atomic force microscope image. K.W., T.J., Z.X., Y.T. and H.L. analyzed the data; the writing of the paper was led by K.W. and T.J. with participation from J.Y., X.Z. and X.C. All authors discussed and commented on the paper.

## Competing interests

The authors declare no competing interests.
