## [Peer Review File · Nature Communications]

Reviewers' comments:

Reviewer #1 (Remarks to the Author):

This is an experimental work which demonstrates the existence of acoustic phonon recycling in a graphene-WS₂ heterostructure. The authors show that acoustic phonons transmitted from the semimetal graphene into the insulator WS₂ can generate conduction electrons in that insulator before they are transmitted into the substrate.

The authors show that a simple thermal transport model is capable of predicting this effect. The theoretical foundations are solid and the arguments and evidence used to show that this is indeed phonon recycling and not some other phenomena, are convincing. Despite the potential impact, there are the following major concerns.

1. To make sure that the phononic excitation is because of graphene phonons, it will need more information on phonon transport (spectral interfacial transmission, etc.) and photonic excitation in WS₂. Since WS₂ is a 2-D material, its properties (excitation, phonon, etc.) are significantly influenced by the spacer or interfacing materials.

2. The graphene and WS₂ contact may not be ideal planar, especially with the inserted h-BN or other spacers [Figure 2(a)]. Acoustic phonons generated in graphene may not be transported if the contact is not sufficiently good.

3. Several figures seem to be missing and some figure captions do not look accurate either.

Figure 1. (d) is missing although the caption is provided and cited in line 93 on page 5. Figure 1 caption title 'Experimental realization and results' are not correct (since 'results' are not provided')
Figure 2- 2(b) and 2(e) are missing. The title of Figure 2 caption contains, 'modeling', but the presented figures are not about modeling.

Figure 4(a) is incomplete [x- and y- axis labels are missing, and it seems to have more figure pane(s)]

Their writing needs improvement:

p2. Line 36: in '... hot carriers to electrical energy...' => I think, 'electrical energy' should be phonon energy,

p.8, line 164: the general exfoliated monolayer WS₂ is n doped' -> with what elements is WS₂ doped?

p. 17: clarify -> clarification

4. It would be more beneficial to provide more guidance how the presented phonon recycling can be applied for high-efficiency electronic and optoelectronic applications.

5. There have been a number of theoretical studies focused on acoustic and optical phonon recycling for the reduction in heat generation or increase in device efficiencies, but very little experimental evidence supporting these efforts, making this a notable and important work. However, It would increase the impact of the study if more connections were drawn to the many theoretical efforts to realize phonon recycling, emphasizing the importance of this work. For Example, Applied Physics Reviews, 6, 021305, 2019; Phys. Rev. B, 96, 205444, 2017; Physical Review B, 91, 85301, 2015; Journal of Applied Physics, 114, 083710-1-9, 2013; Applied Physics Letters, 95, 074103-1-3, 2009.

6. The writing is generally poor, making it difficult to follow to authors in many places. Some Figures also appear to be missing. With minor revision, primarily to the quality of writing, I would recommend that it is published.

Based on the above, the manuscript and its results have merit but not acceptable in the current form. Major revisions are needed and the manuscript should be reviewed again and assessed after revision.

Reviewer #2 (Remarks to the Author):

The authors report a new acoustic phonon recycling (APR) process in graphene-WS2 heterostructures using transient absorption microscopy. APR can couple the lattice heat generated in graphene back into the carrier density in WS2. The APR mechanism is found to have long characteristic time (>100 ps) and broadband response (from around 1.8 to 0.5 eV; below the energy gap of WS2). The authors also find that the APR process can be well described by a thermal transport model at the graphene-WS2 interface. The observation of the TA signal after a long period of excitation in this work is interesting and novel in graphene-WS2 heterostructures. The APR process and relevant heat management are important to the 2D materials community as well as optoelectronic device applications. However, the authors haven't considered the crystal lattice misalignment and energy band mismatch between layers in the heterostructure. The mismatch can affect the heat and charge transport between layers. Some of the experiment details are not clear. Provided the authors address several questions and concerns below, I will consider the manuscript to be published in Nature Communication after revision.

1. The pump photon energy (i.e. 0.48 eV) used in this work is in the sub-bandgap regime, which is far from the WS2 bandgap of 2 eV. Does the selected excitation energy depend on the twist angle between graphene and WS2? Energy band misalignment between graphene and WS2? What are the criteria the authors apply to determine the pump photon energy?

2. The twist angle between layers (graphene-WS2, graphene-hBN, hBN-WS2, WSe2-WS2) should alter the thermal transport between layers, thereby the phononic excitation mentioned in this work. How do the authors manage to account for the effect of layer misalignment on the observation in the manuscript (i.e. Fig. 3)?

3. The authors chose graphene-WS2 heterostructures in this work. However, the APR process seems to work in a different heterostructure consisting of graphene and other transition metal dichalcogenides (TMDs such as WSe2, MoS2). Could the authors explain the motivation of picking up WS2 but not other TMDs?

4. Following Q.3, does bilayer (2L) or even trilayer (3L)-WS2 affect APR in their heterostructures with graphene? It has been reported that the characteristic time of electronic excitation in graphene-2L-WS2 heterostructures is in the same order as in graphene-1L-WS2 heterostructures.

5. As presented in Fig. 2(a), there are many green spots in the heterostructure regions (graphene-WS2 and graphene-hBN-WS2), suggesting poor interfaces between layers. The interface quality in heterostructures should affect the thermal contact resistance and charge transfer between layers. How do the authors rule out the effect of interface quality on the TA feature $\gg 20$ ps?

Reviewer #3 (Remarks to the Author):

The manuscript describes an experimental study on the transient absorption of van der Waals heterostructures made of graphene/WS2 (without and with hBN or WSe2 spacer layers). The authors discuss the charge, energy and heat transfers in this layered system.

In particular, they describe an original signal that appears on a longer timescale than the well studied direct charge/energy transfer of carriers at the interface. They claim that this optical signal is related to a mechanism they refer as "phonon recycling".

Although I support their discussion and modeling for the prior steps of this mechanism: electron thermalization in graphene, thermal transfer to the graphene lattice and subsequently to the WS2

lattice, I am not convinced by the final step: "re-excite the electron-hole pairs in WS₂", which is basically the "phonon recycling" at the heart of the paper.

My claim is that the observed signal can be explained considering only thermal effect in WS₂.

At stake is the discussion at pages 6-7 when the direct lattice heating effect on the transient absorption is ruled out. As it is stated, the experimental signal is very similar to a derivative signal (fig S8) that would originate from spectral shift induced by the increase in temperature only. If it were to be the creation of excitons, the transient would be similar to the signal at short timescale (fig2d or figS5, naming a bleaching at A0 peak). Therefore, I am not convinced by the qualitative and vague arguments used to quickly rule out the direct temperature effect.

I strongly advise the authors to be quantitative here: their model allows to extract the lattice temperature of WS₂ for the experimental fluence, therefore they could simulate the expected change in absorption induced by such heating (using for example data from ref 44, naming the absorption spectra as function of temperature).

I would be happy to see a revised version which includes such quantitative discussion.

At the moment, I would recommend NOT to publish the manuscript in Nature Communications unless this previous point is correctly answered.

If no better evidence for a "phonon recycling" rather than simple lattice heating can be shown, then the manuscript should be heavily modified to present this narrower claim. It would eventually transform into an interesting, well documented, study on thermal management in 2d heterostructures, which might however not be up to the Nature Communications standards.

Response to Reviewers

Reviewer #1:

This is an experimental work which demonstrates the existence of acoustic phonon recycling in a graphene-WS₂ heterostructure. The authors show that acoustic phonons transmitted from the semimetal graphene into the insulator WS₂ can generate conduction electrons in that insulator before they are transmitted into the substrate.

The authors show that a simple thermal transport model is capable of predicting this effect. The theoretical foundations are solid and the arguments and evidence used to show that this is indeed phonon recycling and not some other phenomena, are convincing. Despite the potential impact, there are the following major concerns.

Comment 1: *To make sure that the phononic excitation is because of graphene phonons, it will need more information on phonon transport (spectral interfacial transmission, etc.) and photonic excitation in WS₂. Since WS₂ is a 2-D material, its properties (excitation, phonon, etc.) are significantly influenced by the spacer or interfacing materials.*

Response: We thank the Reviewer for this important comment. Actually, we have shown many experimental results to confirm the phononic excitation in G-WS₂ heterostructure under sub-bandgap excitation (i.e., pump energy smaller than 2 eV). Typical examples include Figure 1d, Figure 2e and f, and Figure 3. To further support this statement, in the new manuscript, we add some experimental evidences from our newly prepared G-WS₂-G sample. Specifically, we verify the APR process through the following steps.

Firstly, we have to ensure that TA signal in the heterostructures is caused by excitons in WS₂ rather than graphene. This can be easily confirmed from the TA spectrum, as shown in Figure 1d and Figure 3 in the main text. Within our detection range, the TA spectra match well with the response of A exciton of WS₂, indicating that the TA signal is completely caused by WS₂ excitons.

Secondly, we should confirm that the WS₂ exciton is originated from graphene rather than directly excited from the pump pulse. This can be better demonstrated from our newly prepared G-WS₂-G sample, as shown in Supplementary Figure 14(a). The CVD grown monolayer WS₂ is inserted between two graphene layers by the same dry transfer process as before, forming the heterostructures containing both the G-WS₂ and G-WS₂-G regions. The PL quenching image shown in Supplementary Figure 14(b) indicates the well interface coupling of all the heterostructure regions. Then, to show the APR process, a femtosecond pulse with 0.48 eV photon energy is used to pump the sample, with a typical TA image from G-WS₂-G region presenting in Supplementary Figure 14(c). Similar to Figure 1(d) in the main text, clear TA features from both electronic and phononic excitations are found, with distinct delay times. Next, to prove that the TA signal is came from Graphene, we carry out a line mapping of TA intensity under two delay times, namely 0.5 and 100 ps, which correspond to electronic and phononic excitations, respectively. As shown in Supplementary Figure 14(c) and 14(d), both the electronic and phononic excitations increase twice (the electronic excitation slightly saturates) when the pump position is moved from G-WS₂ to G-WS₂-G regions. This quantitative result unequivocally shows that both the electronic and phononic excitations are originated from graphene rather than WS₂ or other pathways.

Supplementary Figure 14: APR effect in the G-WS₂-G heterostructure. (a) Optical image of a G-WS₂-G sample. (b) PL intensity mapping showing the well coupling of the heterostructures. (c) 2D pseudo-color TA maps measured at the G-WS₂-G region marked in (a), yellow point. (d-e) Electronic (d) and phononic (e) excitation intensity mapping along the line shown in (a), black dash. (f) A comparison of the TA spectra induced by the electronic and phononic excitations.

Finally, we need to distinguish the electronic and phononic excitations. After confirming that the excited carriers in WS₂ are originated from graphene, the only question is whether the carriers are caused by electronic or phononic excitations. For G-WS₂ heterostructure, the electronic excitation is originated from interfacial carrier transfer, which has been intensively studied in previous reports, with a carrier transfer time of <1 ps. The electronic excitations will completely decay in several ps (typical lifetime of ~1 ps) *via* non-radiative recombination. While for phononic excitation, the characteristic time is much longer since the interfacial thermal transport rate is much slower than that of carrier transfer. According to the experimental data and our simulated result, the peak of the phononic excitation can be longer than 100 ps. This characteristic time varies from sample to sample, determined by several factors, including the heat capacity, the thermal conductance and so on (see Supporting Information **Section 3** for detail discussion). The timescale of 100 ps of the slow TA feature is long enough to rule out the ultrafast electronic excitation. As a result, we attribute the slow TA feature to the phononic excitation from graphene, namely the APR process.

Comment 2: *The graphene and WS₂ contact may not be ideal planar, especially with the inserted h-BN or other spacers [Figure 2(a)]. Acoustic phonons generated in graphene may not be transported if the contact is not sufficiently good.*

Response: We appreciate the Reviewer's insightful concerns. The APR effect is essentially based on the interfacial thermal transport, for which the interlayer thermal conductance (Γ) is closely related to the interface contact. Therefore, it is necessary to explore the dependence of the interface coupling strength on the APR efficiency, since there is no ideal interface in practical devices.

To acquire G-WS₂ heterostructures containing regions with significantly different coupling strength, we prepare new samples using the dry transfer technique similar to the case in the main text but without the final annealing process. The absence of the annealing process remains some

local strains generated during the transfer process, leading to an inhomogeneous interface containing regions with both well and terrible coupling strength. A typical non-annealed sample is shown in Supplementary Figure 10(a), whose interface quality is characterized via PL quenching image under up-bandgap excitation (532 nm). As shown in Supplementary Figure 10(b), regions with both large (well interlayer coupling) and negligible (terrible interlayer coupling) PL quenching are found in the G-WS₂ heterostructures, providing an ideal platform to study the coupling-dependent APR effect. Again, we use a femtosecond pulse with a sub-bandgap photon energy of 0.48 eV to pump the graphene and trace the carrier density excited in WS₂. Since the electronic and phononic excitations dominate the excited carriers at distinct time scales, they can be separately characterized simply by changing the time delays. Supplementary Figure 10(c) shows the TA intensity images induced by the electronic (0.5 ps delay, up panel) and phononic (100 ps delay, below panel) excitations, with the mapping area shown in Supplementary Figure 10(b), solid box. These two TA maps show the same profile as that of PL intensity, indicating that both the electronic and phononic excitations are closely related to the interface quality. Specifically, regions with stronger coupling strength (weaker PL intensity) always show more efficient electronic and phononic excitations. The positive correlation between phononic excitation and interfacial coupling strength can also be predicted by our APR model. An improvement of coupling strength may induce a rise of the interfacial thermal conductance between G and WS₂ layers. In this case, if the heat dissipated rate at WS₂-SiO₂/Si interface does not change, the maximum thermal energy stored in WS₂ at a particular time delay (t_{APR}) will increase, leading to the increase of the excited carriers.

Supplementary Figure 10: Interfacial coupling-dependent APR effect. (a) Optical image of a non-annealed G-WS₂ sample. The heterostructure is marked by the dashed box. (b) PL intensity image showing the coexistence of well and terrible coupling regions. (c) TA intensity images measured at time delay of 0.5 ps (up panel) and 100 ps (below panel), with the mapping area shown in (b), solid rectangle. (d) 2D pseudo-color TA maps of the three heterojunction areas marked in (b). (e) TA spectra of the three heterojunction areas marked in (b), with time delays of 0.5 ps (up panel) and 100 ps (below panel).

For further quantitative study, we choose 3 junctions (Area 1, 2 and 3) where the coupling strength gradually increases for TA kinetics measurement, as marked in the PL and TA images. Supplementary Figure 10(d) shows the TA spectrum relaxations of the 3 areas. Both the electronic and phononic excitations are clearly found at different time scales, i.e., <10 ps and >10 ps, respectively. For comparison, we plot the TA spectra of the 3 areas together, as shown in Supplementary Figure 10(e). Although the electronic and phononic excitations are both positively related to the interface coupling strength, the former is much more sensitive. With coupling strength increasing from area 1 to 3, the electronic excitation increases ~13 times, while the phonon excitation only enhances twice. This can be attributed to different effective transfer ranges of the electron and phonon. Carrier transfer is a short-range process that relies on the orbital overlap between two adjacent layers (~1 nm) or on near-field dipole-dipole coupling (up to several nm), which is extremely sensitive to the interlayer distance. While the phonon transport relies on collective vibration of the crystal lattice, whose effective range is much longer than that of carrier transfer.

Comment 3: *Several figures seem to be missing and some figure captions do not look accurate either.*

Figure 1. (d) is missing although the caption is provided and cited in line 93 on page 5. Figure 1 caption title 'Experimental realization and results' are not correct (since 'results' are not provided)
Figure 2- 2(b) and 2(e) are missing. The title of Figure 2 caption contains, 'modeling', but the presented figures are not about modeling.

*Figure 4(a) is incomplete [x- and y- axis labels are missing, and it seems to have more figure pane(s)]
 Their writing needs improvement:*

p2. Line 36: in '... hot carriers to electrical energy...' => I think, 'electrical energy' should be phonon energy,

p.8, line 164: the general exfoliated monolayer WS₂ is n doped' -> with what elements is WS₂ doped?

p. 17: clarify -> clarification

Response: We thank the Reviewer for the useful suggestions. We are sorry for the format errors and the missing figures during submission of the manuscript. In the new manuscript, we have checked these issues carefully. The corresponding changes have been made:

1. The missing Figure 1(d) has been modified and added. Furthermore, in order to unify the color of the Figures used in the manuscript, all the 2D pseudo-color figures and the mapping figures have been modified using new fill colors.
2. Figure 2(d) and 2(e) have been added. The title of the caption has also been modified to "APR process induced by the rising Fermi energy" to match the entire figure.
3. The labels of the Figure 4(a) have been added. Figure 4 only contains three panels, a, b and c, which characterizes the APR properties under different pump power. Part of the caption has also been modified.
4. Actually, what we mean is that to prevent the conversion of the hot carrier energy to optical phonon, the carrier must reach the electrode within 0.2 ps after excitation, and this is really challenging even for graphene device with an ultrahigh carrier mobility. As a result, most of the hot carrier energy will be transferred to phonons and finally wasted as heat. For

clarification, we change the statement as “Thus, to acquire a device with both high performance and low heat generation, the excited hot carriers must be collected by the electrode within 0.2 ps before their energy is transferred to the optical phonons”.

5. Generally, the as-prepared WS₂ or MoS₂ monolayer is slightly *n* dope due to sulfur vacancies or/and doping from SiO₂/Si substrate. This has been added in the new manuscript.
6. The wrong word ‘clarify’ has been rectified.

Comment 4: *It would be more beneficial to provide more guidance how the presented phonon recycling can be applied for high-efficiency electronic and optoelectronic applications.*

Response: We thank the Reviewer for the constructive suggestion. Since the APR effect has already transferred the heat generated in graphene to the charge carrier distribution in WS₂, so we think the following energy collection of this phonon excited carriers is simple, just as the configuration proposed by Koppens group (Massicotte, M. et al. *Nat. Commun.* 7, 12174 (2016)). In that case they design a similar graphene-based (both G-WSe₂ and G-WSe₂-G) field-effect transistor (FET) to explore the photo-thermionic effect, which is ascribed to the electronic excitation in our paper since the interlayer transfer process is mediated by the photocarriers. The generated photocurrent in that paper was completely attributed to the photo-thermionic effect, or electronic excitation. While based on our study here, we propose that the APR effect cannot be ignored in similar graphene-based devices. Specifically, as shown in Figure 4(c), we can find that although the peak amplitude of the APR effect is ~13 times smaller than that of the carrier transfer, the steady and prolonged heat flows may make a considerable contributions to the electrical response of the device. If we consider the temporal integration of the reflectance signal, APR will play a more important role (i.e., ~50 times) than the ultrafast carrier transfer process.

Comment 5: *There have been a number of theoretical studies focused on acoustic and optical phonon recycling for the reduction in heat generation or increase in device efficiencies, but very little experimental evidence supporting these efforts, making this a notable and important work. However, It would increase the impact of the study if more connections were drawn to the many theoretical efforts to realize phonon recycling, emphasizing the importance of this work. For Example, Applied Physics Reviews, 6, 021305, 2019; Phys. Rev. B, 96, 205444, 2017; Physical Review B, 91, 85301, 2015; Journal of Applied Physics, 114, 083710-1-9, 2013; Applied Physics Letters, 95, 074103-1-3, 2009.*

Response: Thank the Reviewer for the recommended papers. We are very happy to see so many theoretical works concentrating on the harvesting of the phonons. Since the discovery of graphene (G), most G-based electronic and optoelectronic devices are concentrated on the charge carrier transport. But the fact is, as we discussed in the main text, charge carriers only have 0.2 ps to move to the electrode before they lose their energy as heat, which is extremely difficult in practical device. As a result, most of the absorbed energy is lost, and that’s why we were trying to do this work. By building heterostructures, we experimentally found that the lost energy in G can be transferred to monolayer WS₂ and carriers are further excited in the letter. This re-excitation process means that we can indeed recycle the wasted phonon energy by simply adding an electric field to the WS₂ to separate the electron and hole excited by the APR effect. Despite the experimentally demonstrating APR process, the theoretical work mentioned above indeed makes our study more systematic and interesting to the community. Thus, these theoretical works are appropriately cited in our new

Reviewer #2:

The authors report a new acoustic phonon recycling (APR) process in graphene-WS₂ heterostructures using transient absorption microscopy. APR can couple the lattice heat generated in graphene back into the carrier density in WS₂. The APR mechanism is found to have long characteristic time (>100 ps) and broadband response (from around 1.8 to 0.5 eV; below the energy gap of WS₂). The authors also find that the APR process can be well described by a thermal transport model at the graphene-WS₂ interface. The observation of the TA signal after a long period of excitation in this work is interesting and novel in graphene-WS₂ heterostructures. The APR process and relevant heat management are important to the 2D materials community as well as optoelectronic device applications. However, the authors haven't considered the crystal lattice misalignment and energy band mismatch between layers in the heterostructure. The mismatch can affect the heat and charge transport between layers. Some of the experiment details are not clear. Provided the authors address several questions and concerns below, I will consider the manuscript to be published in Nature Communication after revision.

Comment 1: *The pump photon energy (i.e. 0.48 eV) used in this work is in the sub-bandgap regime, which is far from the WS₂ bandgap of 2 eV. Does the selected excitation energy depend on the twist angle between graphene and WS₂? Energy band misalignment between graphene and WS₂? What are the criteria the authors apply to determine the pump photon energy?*

Comment 2: *The twist angle between layers (graphene-WS₂, graphene-hBN, hBN-WS₂, WSe₂-WS₂) should alter the thermal transport between layers, thereby the phononic excitation mentioned in this work. How do the authors manage to account for the effect of layer misalignment on the observation in the manuscript (i.e. Fig. 3)?*

Response: We thank the Reviewer for these constructive comments. (1) **Selection of the pump photon energy.** As we mentioned in the main text, we choose two types of pump energy to excite the heterostructure. One is 'up bandgap' pumping, which uses a photon with energy larger than the WS₂ optical bandgap to excite the carriers. This high energy photon simultaneously excites the carriers in WS₂ and graphene, but the case of WS₂ is dominant as it has a much larger absorption coefficient than graphene. Thus, the following processes are that charges are separated and carrier transfer from WS₂ to graphene, leading to the efficient PL quenching of the former. The other excitation condition is sub-bandgap pumping, which uses a pump photon with energy lower than the WS₂ optical bandgap to excite the heterostructure. In this case, carriers can only be pumped at graphene layer since our pump intensity is much lower than the two-photon absorption threshold. Thus, the detected WS₂ carriers are came from interlayer transfer from graphene. For this transfer, we found two excitation peaks with distinct time scales. The fast one peaks at ~0.5 ps and is attributed to interlayer carrier transfer, or electronic excitation here. The slow excitation peak occurs at a much longer delay time, with a peaking time of >100 ps, we attributed this slow feature to acoustic phonon recycling (APR), or phononic excitation, a similar concept to the electronic excitation but occurring via interlayer thermal transport. This new APR mechanism is not only found in a particular photon energy (e.g. 0.48 eV used in the main text), but commonly found in all sub-bandgap pumping cases (pump photon energy < 2 eV, see Supplementary Figure 15). In this study, limiting by the femtosecond light sources, we only carried out experiments with photon energy down to 0.48 eV (2600 nm). However, we believe that this APR effect (and also the electronic

excitation) can maintain to a much lower pump energy.

(2) **Dependence of twist angle on APR effect.** Since the conduction band minimum and the valance band maximum of the monolayer transition metal dichalcogenides (TMDCs) are both located at K point in the momentum space, interfacial charge transfer process between graphene (G) and WS₂ involves change in parallel momentum vector once their twist angle is not zero. Moreover, the band alignment between the G and WS₂ monolayer also changes with varying twist angles. As a result, general belief and theoretical studies have shown that the interfacial charge transfer processes, including both the transfer rate and efficiency, depend sensitively on the *twist angle* (ϕ) between two adjacent layers. Therefore, as the reviewer's comment here, it is very important and urgent to study the effect of twist angle and energy band misalignment on the acoustic phonon recycling (APR) process in G-WS₂ heterostructures. Base on this we prepare new heterostructure samples and carry out related experiments, with the results showing in the supporting information Section 6 of the new manuscript and also listing below.

The twist angle ϕ is defined as the angle between the zigzag (or armchair) directions of the G and WS₂ layers, where ϕ can vary from 0-30° considering the 6-fold symmetry of the G lattice. To eliminate the diversity from sample to sample, we choose the monolayer WS₂ layer grown by chemical vapor deposition (CVD) rather than mechanical exfoliation, which contains WS₂ triangles with different crystal orientations in one sample. Then a mechanical exfoliated graphene flake was transferred onto the CVD WS₂ film, forming heterostructures with different twist angles. The orientation of graphene is determined by Raman spectroscopy, while the orientation of WS₂ triangle is identified from the optical image, with three sides orienting at zigzag direction.

A typical G-WS₂ heterostructure on SiO₂/Si substrate is shown in Supplementary Figure 11(a). A femtosecond laser pulse (0.48 eV, ~34 μ J/cm²) is used to excite the heterostructure to show the APR process. Supplementary Figure 11(b) shows a typical relaxation kinetics of A exciton excited in the heterostructure with $\phi=1.5^\circ$. Although the secondly rise peak is not so obvious from naked eyes compared to the samples used in the main text, we can still see a nearly flat platform from 20 to ~100 ps, which is caused by the APR effect. Next, to determine the dependence of phononic excitation and angular alignment, we carry out TA intensity mapping at two fixed delay times, 0.5 and 100 ps, as shown in Supplementary Figure 11(c) and (d). Three heterostructure configurations are shown in the two images, with $\phi = 1.5^\circ, 23^\circ$ and 30° . Within the range of noise, both the electronic (0.5 ps) and phononic (100 ps) excitation intensity show no correlation with the twist angle. The TA spectra caused by these two excitations also show no relationship with the angular alignment (Supplementary Figure 11(e)). To further give a quantitative comparison, we made a statistic of the electronic and phononic excitations from 11 configurations in this sample, with data of each configuration averaging from 10 different regions. As shown in Supplementary Figure 11(f), again, there is no obvious dependence between electronic (phononic) excitation and twist angle.

The absence of correlation between electronic (phononic) excitation and ϕ seems quite counterintuitive and is also not consistent with some previous theory studies. However, we note that many experimental studies also show similar result in different types of TMDC heterostructures, including MoS₂-WSe₂, WS₂-MoS₂ and MoS₂-MoSe₂. The interlayer charge transfer is found to be extremely robust against varying interlayer twist angles. Two mechanisms are proposed accounting for this robust carrier transfer. The first one is hot carrier transfer, i.e. the excess kinetic energy of the transferred carriers allows them to sample a broad range of K space above the CBM (K point). The second mechanism is the coexistence of various local stacking configurations caused by

atomically local structural inhomogeneity during the sample preparation. The lattice constant difference of the graphene (2.46 Å) and WS₂ (3.15 Å) will introduce local strain when the two layers match together. Then in the following annealing process, the local strain will be further amplified and bring interlayer shift due to the different thermal expansion coefficients of the two layers. The interlayer sliding results in various local stacking configurations and thus carrier may transfer through multichannels.

Although these two mechanisms are proposed for carrier transfer, i.e. electronic excitation here, similar concept can be applied to the phononic excitation. The existent of hot phonon and the heterogeneous interlayer stretching/sliding may also induce a robust phononic excitation against varying twist angle between the two monolayer components.

Supplementary Figure 11. Stacking-independent APR effect. (a) Optical image of a typical G-WS₂ sample with different stacking configurations. (b) Carrier relaxation dynamics in a G-WS₂ heterostructure with $\phi = 1.5^\circ$. (c-d) TA intensity images induced by electronic (c) and phononic (d) excitations, measured at delays of 0.5 ps and 100 ps, respectively. (e) TA spectra caused by electronic (up panel) and phononic (below panel) excitations from three stacking configurations ($\phi = 1.5^\circ, 23^\circ$ and 30°). (f) Electronic (up panel) and phononic (below panel) excitation intensities versus twist angle of graphene and WS₂. No correlation is found between these two excitations and twist angle.

Comment 3: The authors chose graphene-WS₂ heterostructures in this work. However, the APR process seems to work in a different heterostructure consisting of graphene and other transition metal dichalcogenides (TMDs such as WSe₂, MoS₂). Could the authors explain the motivation of picking up WS but not other TMDs?

Response: We thank the Reviewer for the important concern. Actually, we are free to choose the materials. The reason why we choose WS₂ is that WS₂ has the highest PL quantum yields in TMDs, which is commonly used in our previous works. But when it comes to the APR process, this new effect found in our manuscript, we believe that all the TMDs can support such an interfacial thermal transport process. However, there are some differences for different TMDs.

$$\Delta R(t) / R_0 \propto -\Delta n(t) / n_s \cong \frac{N_c (E_c - E_F) \Delta T(t)}{n_s k_B T_0^2} \exp\left(-\frac{E_c - E_F}{k_B T_0}\right) \quad (1)$$

According to our model (i.e., equation 1 in the main text), the detected differential reflection intensity is proportional to two key factors, namely the Fermi energy E_F and the lattice temperature increase $\Delta T(t)$. The first factor is dependent on the doping level of the material, while the second one is related to the thermal capacity and the interfacial coupling strength (including graphene-WS₂ and WS₂-Si/SiO₂ interfaces). For graphene-based heterostructures with different TMDs monolayers, these two variables are completely different, especially for E_F . From figure 2(d) in the main text, we can find that the optical property (both PL emission and transient absorption) of WS₂ in the heterostructure region is completely dominated by charge exciton (A⁻), indicating that the WS₂ fermi energy is greatly enhanced during sample preparation, which is favorable for the APR process according to equation 1.

Supplementary Figure 13. APR effect in G-MoS₂. (a) optical image of a typical G - nL WS₂ sample. (b) 2D pseudo-color TA maps of different sample regions with different pump energy. Left, controlled MoS₂, $E_{\text{pump}}=2.1$ eV, middle, G-MoS₂, $E_{\text{pump}}=2.1$ eV and right, G-MoS₂, $E_{\text{pump}}=0.48$ eV. (c) Corresponding carrier relaxation dynamics. (d) TA spectra of the G-MoS₂ at different time delays, with pump photon energy of 0.48 eV. (e) The comparison of TA spectra from different sample regions. All the TA spectra are dominated by neutral excitons (A⁰ and B⁰), indicating an intrinsic doping level of the MoS₂ layer.

However, this enhanced E_F is not found in our new as-prepared graphene-MoS₂ heterostructure (Supplementary Figure 13(a)). Specifically, the TA spectra measured in the controlled MoS₂ region and the heterostructure region (Supplementary Figure 13(e)) are completely the same, all dominated by the neutral excitons (A⁰ and B⁰). This indicates that E_F locates at an energy level near the neutral point, far from the conduction band minimum E_C . As a result, we cannot see any APR process in this sample. As shown in Supplementary Figure 13(b) and 13(c), the carrier density increases quickly in the first 0.5 ps, which is caused by the interfacial carrier transfer rather than the APR process. Then in the following delay time, the carriers density decrease monotonously at the whole detection range (0~1.4 ns), no matter what photon energy is used for light excitation (2.1 eV and 0.48 eV). The shape of the TA spectra excited by 0.48 eV femtosecond pulse is also the same at all the delay times (Supplementary Figure 13(d)), further indicating that interfacial carrier transfer is the only mechanism accounting for the excited carriers in MoS₂.

Note that although the APR process is not found in the as-prepared G-MoS₂ heterostructure, if we can raise the Fermi level of MoS₂ from intrinsic state to an energy level near the E_C , just as the case for G-WS₂, we believe that the APR process will appear. Actually, the APR efficiency is also distinct in different G-WS₂ samples (see Figure 4(a), Figure 4(c) and Supplementary Figure 14), possibly caused by the change of Fermi energy. In the main text we have a simple estimate of the influence of E_C on the detection signal $\Delta R / R_0$, a 100 meV rise of Fermi energy leads to a dramatic increase of $\Delta R / R_0$ from 0.001 to 0.027. Thus, electrical gate control may be an efficient strategy to improve the APR efficiency, and that is exactly what we will do next.

Comment 4: *Following Q.3, does bilayer (2L) or even trilayer (3L)-WS₂ affect APR in their heterostructures with graphene? It has been reported that the characteristic time of electronic excitation in graphene-2L-WS₂ heterostructures is in the same order as in graphene-1L-WS₂ heterostructures.*

Response: We thank the Reviewer for this useful comment. Similar to the G-MoS₂ sample mentioned above, we first have a simple analysis on the difference between monolayer and few layer WS₂. If we pick up the WS₂ sample with different layers from one bulk single crystal and then transfer one graphene flake on them, the only different between monolayer and few layer WS₂ is heat capacity for interfacial thermal transport, the origin of APR effect we discussed here. The heat capacity of WS₂ is approximately proportional to the number of layers, i.e., $C_{WS_2(n)} = n C_{WS_2(1)}$, where $C_{WS_2(1)} = 4.5 \times 10^{-4} \text{ J} \cdot \text{m}^{-2} \cdot \text{K}^{-1}$ is the monolayer heat capacity. Since the total energy stored in the G-WS₂ heterojunction system is constant under a fixed pump intensity, the increase of C_{WS_2} will naturally lead to the decrease of lattice temperature rise ΔT_{WS_2} , and thus a decrease of the APR efficiency.

Furthermore, the increase of WS₂ layer will significantly increase the peak delay time of APR-induced photo carriers (t_{APR}) because it takes more time from thermal transport and balance. Based on the thermal transport model described in the main text, we simulate the dependence of ΔT_{WS_2} (t_{APR}) on the WS₂ layers (n), as shown in Supplementary Figure 12a(b).

Next, to confirm our model, we prepare G-WS₂ heterostructure with mechanically exfoliated graphene and n L WS₂, as shown in Supplementary Figure 12(c). Pumping with 0.48 eV femtosecond laser pulse, the 2D pseudo-color TA maps of three heterostructure regions are shown in Supplementary Figure 12(d), namely, G- 1L WS₂, G- 2L WS₂ and G- bulk WS₂. Although not as obvious as the sample used in the main text, all 3 regions show slightly APR effect, with characteristic time (t_{APR}) showing by vertical dashed line. These experimental t_{APR} values are compared with the simulated ones in Supplementary Figure 12(b), which exhibits excellent agreement, again confirming the APR process. Note that with increasing WS₂ layers ($n \geq 2$), the relaxation of the carriers (the PB peak) pump by the initial electronic excitation are greatly slowed down, which may seriously hinder the observation of the following slow APR effect. But from the kinetics of the PIA peaks (~ 2.0 eV for $n = 2$ layer and ~ 1.95 eV for bulk WS₂), we can clearly find the secondly rise feature resulting from the APR effect.

Supplementary Figure 12. APR effect in G-WS₂ with different WS₂ layers. (a-b) Simulated WS₂ layer dependent maximum temperature increase (a) and APR characteristic time (b) under a pump intensity of $\sim 34 \mu\text{J}/\text{cm}^2$. For comparison, Experimental results of the t_{APR} is also plotted at (b), red circles. (c) Optical image of a typical G - n L WS₂ sample. (d) 2D pseudo-color TA maps of different heterostructure regions, including G- 1L WS₂, G- 2L WS₂ and G- bulk WS₂, the APR characteristic time is marked by the dash line.

Comment 5: As presented in Fig. 2(a), there are many green spots in the heterostructure regions (graphene-WS₂ and graphene-hBN-WS₂), suggesting poor interfaces between layers. The interface quality in heterostructures should affect the thermal contact resistance and charge transfer between layers. How do the authors rule out the effect of interface quality on the TA feature $\gg 20$ ps?

Response: Many thanks for this important comments. The APR effect is essentially based on the interfacial thermal transport, for which the interlayer thermal conductance (Γ) is closely related to the interface contact. Thus, it is necessary to explore the dependence of the interface coupling strength on the APR efficiency, as there is no ideal interface in practical devices.

To acquire G-WS₂ heterostructures containing regions with significantly different coupling strength, we prepare new samples using the dry transfer technique similar to the case of the main text but without the final annealing process. The absence of the annealing process remains some local strains generated during the transfer process, leading to an inhomogeneous interface containing both well and terrible coupling strengths. A typical non-annealed sample is shown in Supplementary Figure 10(a), whose interface quality is characterized via PL quenching image under up-bandgap excitation (532 nm). As shown in Supplementary Figure 10(b), regions with both large (well interlayer coupling) and negligible (terrible interlayer coupling) PL quenching are found in the G-WS₂ heterostructures, providing an ideal platform to study the coupling-dependent APR effect. Again, we use a femtosecond pulse with a sub-bandgap photon energy of 0.48 eV to pump the graphene and trace the carrier density excited in WS₂. Since the electronic and phononic excitations dominate the excited carriers at distinct time scales, they can be separately characterized simply by changing the time delays. Supplementary Figure 10(c) show the TA intensity images induced by the electronic (0.5 ps delay, up panel) and phononic (100 ps delay, below panel) excitations, with the mapping area shown in Supplementary Figure 10(b), solid box. Both the TA maps show the same profile as that of PL intensity, indicating that both the electronic and phononic excitations are closely related to the interface quality. Specifically, the weaker PL signal, the stronger coupling strength and thus the more efficient electronic and phononic excitations. The positive correlation between phononic excitation and interfacial coupling strength can also be predicted from our APR model. An improvement of coupling strength may induce a rise of the interfacial thermal conductance between G and WS₂ layers. In this case, if the heat dissipated rate at WS₂-SiO₂/Si interface does not change, the maximum thermal energy stored in WS₂ at a particular time delay (t_{APR}) will increase, leading to the increase of the excited carriers.

For further quantitative study, we choose 3 junctions (Area 1, 2 and 3) where the coupling strength gradually increases for TA kinetics measurement, as marked in the PL and TA images. Supplementary Figure 10(d) shows the TA spectrum relaxation of the 3 areas. Both the electronic and phononic excitations are clearly found at different time scales, i.e. <10 ps and >10 ps, respectively. For comparison, we plot the TA spectra of the 3 areas together, as shown in Supplementary Figure 10(e), for both electronic (up panel) and phononic (below panel) excitations. Although these two excitations are both positively related to the interface coupling strength, the electronic excitation is much more sensitive. With coupling strength increasing from area 1 to 3, the carrier transfer induced electronic excitation increases ~ 13 time, while the phonon excitation only enhances twice. This can be attributed to different effective transfer ranges of the electron and phonon. Carrier transfer is a short-range process that relies on the orbital overlap between two adjacent layers (~ 1 nm) or on near-field dipole-dipole coupling (up to several nm), which is

Supplementary Figure 10. Interfacial coupling-dependent APR effect. (a) Optical image of a non-annealed G-WS₂ heterostructure. (b) Corresponding PL intensity image. (c) Corresponding TA intensity images measured at time delay of 0.5 ps (up panel) and 100 ps (below panel). (d) 2D pseudo-color TA maps of the three heterojunction areas marked in (b). (e) TA spectrum of the three heterojunction areas marked in (b), with time delay of 0.5 ps (up panel) and 100 ps (below panel).

extremely sensitive to the interlayer distance. While the phonon transport relies on collective vibration of the crystal lattice, whose effective range is much longer than that of carrier transfer.

Reviewer #3:

The manuscript describes an experimental study on the transient absorption of van der Waals heterostructures made of graphene/WS₂ (without and with hBN or WSe₂ spacer layers). The authors discuss the charge, energy and heat transfers in this layered system.

In particular, they describe an original signal that appears on a longer timescale than the well studied direct charge/energy transfer of carriers at the interface. They claim that this optical signal is related to a mechanism they refer as “phonon recycling”.

Although I support their discussion and modeling for the prior steps of this mechanism: electron thermalization in graphene, thermal transfer to the graphene lattice and subsequently to the WS₂ lattice, I am not convinced by the final step: “re-excite the electron-hole pairs in WS₂”, which is basically the “phonon recycling” at the heart of the paper.

My claim is that the observed signal can be explained considering only thermal effect in WS₂. At stake is the discussion at pages 6-7 when the direct lattice heating effect on the transient absorption is ruled out. As it is stated, the experimental signal is very similar to a derivative signal (fig S8) that would originate from spectral shift induced by the increase in temperature only. If it were to be the creation of excitons, the transient would be similar to the signal at short timescale (fig2d or figS5, naming a bleaching at A0 peak). Therefore, I am not convinced by the qualitative and vague arguments used to quickly rule out the direct temperature effect.

I strongly advise the authors to be quantitative here: their model allows to extract the lattice temperature of WS₂ for the experimental fluence, therefore they could simulate the expected change in absorption induced by such heating (using for example data from ref 44, naming the absorption spectra as function of temperature).

I would be happy to see a revised version which includes such quantitative discussion. At the moment, I would recommend NOT to publish the manuscript in Nature Communications unless this previous point is correctly answered.

If no better evidence for a “phonon recycling” rather than simple lattice heating can be shown, then the manuscript should be heavily modified to present this narrower claim. It would eventually transform into an interesting, well documented, study on thermal management in 2d heterostructures, which might however not be up to the Nature Communications standards.

Response: Many thanks for the Reviewer for such insightful and constructive comments. What the reviewer confuses is that whether the TA signal at a time scale much larger than 10 ps is originated from the pure crystal heating or the phonon-excited carriers (acoustic phonon recycling). In the original manuscript, we only have a brief discussion on the crystal heating effect, which is too simple and qualitative. Now in the new manuscript (**Page 7** in the main text, and Supporting information **Section 4**), we carry out an in-depth discussion on the origination of the PB and PIA features in TA spectrum of the general TMDCs. Moreover, according to the reviewer’s comments, we also add a quantitative study on the dependence of the decreased optical bandgap on the increasing crystal temperature. Based on these analyses, we can clearly point out that the derivative TA signal is caused by the phonon-excited carriers rather than the pure crystal heating effect. For the sake of discussion, we summarize as following:

Firstly, although the TA signal at time delay much larger than 10 ps looks like a derivative feature, it is not a characteristic for the phononic excitation. Instead, the derivative feature may also occur at delay time of < 10 ps in some samples, such as the Sample 2 used in our main text (Fig. 3c in the

main text, and also shown in Fig r1(b) and r1(d) below), during which electronic excitation dominates. Thus, simply from a derivative TA signal, we cannot distinguish the crystal heating from the phonon-excited carriers. We need to have more in-depth analyses.

Figure r1. TA spectra caused by electronic excitation in different G-WS₂ samples. (a) 2D pseudo-color TA maps of the G/WS₂-based heterostructure (Sample 1) with (bottom) and without (up) *h*-BN spacer. (b) 2D pseudo-color TA maps of the G/WS₂-based heterostructure (Sample 2) with (bottom) and without (up) monolayer WSe₂ spacer. (c) TA spectrum taken at the peak delay time (~0.5 ps) of the electronic excitation in G-WS₂ region in Sample 1 (dash line (a)). (d) TA spectrum taken at the peak delay time (~0.5 ps) of the electronic excitation in G-WS₂ region in Sample 2 (dash line (b)). Note that a derivative feature is found in the TA spectrum of Sample 2.

Secondly, a detailed discussion on the crystal heating effect is provided here. Upon sub-bandgap excitation, the excited carriers in graphene (only graphene can be pumped) transfer most of their energy to the graphene lattice via phonon emission, and then to the WS₂ lattice via interfacial thermal transport. The heating of the WS₂ lattice may induce a red shift of the exciton resonance (optical bandgap) due to electron-phonon coupling. According to our previous study (*Appl. Optics* **55**, 6251-6255 (2016).), the temperature-dependent exciton resonance follows the equation below:

$$E(T) = E(0) - S \cdot \hbar\omega \left[\coth(\hbar\omega / 2k_B T) - 1 \right]$$

where $E(T)$ is the exciton resonance at a select temperature of T , $S = 2.4$ is dimensionless coupling strength, $\hbar\omega = 31$ meV is the phonon energy. At room temperature of 300 K, this equation gives a shift rate of ~0.37 meV/K. This value is exactly the same with other previous studies (0.26~0.5 meV/K, most values are closer to 0.3 meV/K). Thus, for the pump intensity of ~34 μJ/cm² used here, the maximum lattice temperature rise of WS₂ is estimated to be ~11 K, giving a band shift of $\Delta E_p \sim 4$ meV. This shift amplitude is much smaller than the energy gap between the PIA and PB features in the TA spectrum, which is estimated to be 25~34 meV (Supplementary Figure 7),

differing from sample to sample. Thus, such a large energy gap between the PIA and PB peaks allows us to safely exclude the pure crystal heating effect. We also note that there are several experimental studies showing the exciton band shift caused by the lattice heating effect, which is determined to be several meV depending on the pump fluence (*Nano Lett.* **17**, 644-651 (2017); *ACS Nano* **11**, 12601-12608 (2017)), comparable well with the estimated value above.

The crystal heating effect can be further eliminated through the pump fluence-dependent TA spectrum measurements, as shown in Supplementary Figure 7. For a pump power range from 11~91 $\mu\text{J}/\text{cm}^2$, which corresponds to a temperature increase of 3.7~29 K and a band red shift of 1.4~10.8 meV, no sizable PIA and PB peak shifts are found in both G-WS₂ and G-*h*-BN-WS₂ regions. This indicates that in our pump power range, the pure crystal heating effect is small enough to be concealed by the influence from phonon-excited carriers.

Supplementary Figure 7: TA spectrum caused by phononic excitation. (a) Pump power dependence on the TA spectrum of the graphene-WS₂ region. (b) Pump power dependence on the TA spectrum of the graphene-*h*-BN-WS₂ region. The red and blue dots show the peak energy of the PIA and PB features, which is clearly independent on the pump power within a range of 11~91 $\mu\text{J}/\text{cm}^2$. The energy difference between the PIA and PB features is ~25 meV and ~34 meV in the graphene-WS₂ and graphene-*h*-BN-WS₂ region, respectively.

Thirdly, we would like to have a discussion on the influence of the phonon-excited carriers on the TA spectrum, or the origin of the PIA feature in TMDCs. In 2D (monolayer or few layers) TMDCs family, it is commonly found that **many-body effect** plays an important role in the optical response, that is, the existing photocarriers excited by previous laser pulse alters the optical absorption of the latter-coming laser pulse. When it comes to the pump-probe measurement, many-body effect appears as a change of the probe absorption in the presence of the pump pulse. Several distinct physical processes contribute to this absorption change. (1) Phase-space filling (or Pauli-blocking) and Coulomb scattering of the carrier result in a decrease oscillator strength and spectral broadening, as presented in Supplementary Figure 9(a). These two effects lead to a PB peak at the exciton resonance and simultaneously a small PIA feature at the energy nearby. (2) Screening of the Coulomb attraction leads to a reduced exciton binding energy and thus a blue shift of the A exciton resonance, as shown in Supplementary Figure 9(b). This effect induces a PB peak at the primary band and a PIA peak at the new blue shift band. (3) Screening of the Coulomb repulsion induce a reduced quasi-particle bandgap (or electronic bandgap) and thus a red shift of the A exciton resonance, as shown in Supplementary Figure 9(c). The decrease of the exciton resonance induces a corresponding PIA feature at the red shift band and a PB peak at the originated band. Note that the

last two processes have opposite effect on the A exciton resonance, thus the band energy either shifts to higher or lower energy. Moreover, these two processes partially compensate each other and give rise to an overall peak shift no more than a few tens millielectronvolts. The shift direction as well as the shift amplitude depends on a variety of material properties and excitation conditions, including the effective temperature and density of the photoexcited carriers, as well as the ratio of the excitons to free electrons and holes. Thus, it is reasonable that different WS₂ samples show different TA spectra under the same electronic excitation (Supplementary Figure 8). Or, conversely, the same sample shows different TA spectra under electronic and phononic excitations. All these variations are due to the complicated many-body effect in the TMDCs.

Supplementary Figure 8: Transition absorption spectra in different samples. (a-c) TA spectra induced by electronic excitation in Sample 1~3. (d-e) TA spectra induced by phononic excitation in Sample 1~3. The PB and PIA feature are marked by the blue and red areas. Note that both the amplitude and energy of the PIA feature vary from sample to sample compared to the PB peak.

Supplementary Figure 9. TA spectrum induced by different kinds of many-body effects. (a) Phase-space filling and Coulomb scattering induced photo bleaching and spectral broadening. (b) Screening of the Coulomb attraction induce the decrease of exciton binding energy (E_b). (c) Screening of the Coulomb repulsion induce the decrease of electronic bandgap (E_g). The up panel illuminates the different kinds of many-body effects and the below panel shows corresponding TA spectra.

Based on the discussion above, we believe that that TA signal at time delay of >10 ps is mainly

attributed to the phonon-excited carriers (acoustic phonon recycling) rather than the crystal heating effect.

Reviewer #1 (Remarks to the Author):

The composite layers although not allowing for a more direct observation of the phonon recycling and its harvesting, is a challenging first step in that direction.

The authors have addressed the comments of this reviewer in a relatively satisfactory manner. So, the publication is recommended.

Reviewer #2 (Remarks to the Author):

In the revised manuscript, the authors have carefully addressed my concerns with additional experiments and calculations. Although there are some unexpected new findings on the dependence of the APR effect on twist angle between WS₂ and graphene as well as no observable APR in graphene-MoS₂ heterostructures, the authors explain with good reasoning. According to the revision, I would like to recommend this paper for publication.

Reviewer #3 (Remarks to the Author):

I kindly thank the authors for their in-depth response.

As well-understood and stated by the authors, my concern is “whether the TA signal at a time scale much larger than 10 ps is originated from the pure crystal heating or the phonon-excited carriers (acoustic phonon recycling)”, which directly relates to the main claim and originality of the manuscript.

Here I maintain my skepticism. I still believe that the TA signal they observe can be explained with pure crystal heating. In fact, using new information they provide, I will quantitatively prove my point. In the following I will review the 3 points the authors make in their response:

POINT 1: I completely agree on their claim that a derivative feature in the TA spectrum is not a direct proof that crystal heating is occurring but can be originated by excited charge carriers too. Therefore, as they stated, “We need to have more in-depth analyses”.

POINT 2: The authors make here a fundamental mistake when describing the effect that a spectral shift due to crystal heating would have. Indeed, for a small spectral shift, we get a TA spectrum that has the form of a derivative where : A) the spectral distance between the PB and PIA peaks relates to the width of the original absorption peak and B) the amplitude of the PB and PIA peaks relates to the amplitude of the spectral shift of the original absorption peak. The case where spectral distance between PB and PIA relates to the spectral shift is for shifts larger than the linewidth: this is not the case here and this would yield TA amplitudes close to unity (not the case here neither). Therefore, considering the data they provide (in Supplementary Figure 7a), we get from A) that the original absorption peak has a width around 20-35 meV, which is the typical width they observe in PL for the exciton lines (Figure2), and from B) that the spectral shift increases almost linearly with pump fluence, which is expected for crystal heating (with a coefficient around -0.3meV/K). Below is a quick modeling I did to better prove my point. I calculated TA spectra originating from a Lorentzian line (width of 20 meV, dashed curve) with different spectral shifts (noted dE). The typical data they provide can be quantitatively reproduced. Here the amplitude of the spectral shifts is below -0.05 meV, which corresponds to a rise in temperature below 0.2K. This lattice temperature rise is even lower than the one the authors claim to reach (up to 30K). This therefore points out another inconsistency in the overall work that needs to be commented (it can be compared to ref Nano Lett. 2017, 17, 2, 644-651, under similar conditions tens of K are reached only at short timescale and only few K or less remain after hundreds of ps).

POINT 3: Here I agree with the authors that many shapes for TA spectra can be obtain when

considering the electronic excitation, due to the large sensitivity with the materials conditions of the many-body effects. This corresponds well to the short timescale signals. Yet I point out that the short timescale signals have more variations than the long timescale signals. The latter always present a derivative shape, with a PIA at lower energy than the PB. This is compatible with pure crystal heating (see point 2) and I guess that even better matching could be obtained if broadening of the line due to heating, as well as the evolution (spectral shift and broadening) of other exciton lines (neutral and charged), would be considered.

From this (now quantitative) discussion, a TA signal at long timescale originating simply from pure lattice heating cannot be ruled out and would correspond to the simplest explanation. The main claim about the observation of APR is therefore not sufficiently supported.

Considering that my main concern is still open (and even strengthened with now a quantitative analysis), I maintain my former opinion: I would recommend NOT to publish the manuscript in Nature Communications. Since no better evidence for a "phonon recycling" rather than simple lattice heating has been shown, then the manuscript should be heavily modified to present this narrower claim. It would eventually transform into an interesting, well documented, study on thermal management in 2d heterostructures. I would then be glad to review such revised manuscript.

Response to Reviewers

Reviewer #3:

Comment: *I kindly thank the authors for their in-depth response.*

As well-understood and stated by the authors, my concern is “whether the TA signal at a time scale much larger than 10 ps is originated from the pure crystal heating or the phonon-excited carriers (acoustic phonon recycling)”, which directly relates to the main claim and originality of the manuscript.

Here I maintain my skepticism. I still believe that the TA signal they observe can be explained with pure crystal heating. In fact, using new information they provide, I will quantitatively prove my point. In the following I will review the 3 points the authors make in their response:

POINT 1: I completely agree on their claim that a derivative feature in the TA spectrum is not a direct proof that crystal heating is occurring but can be originated by excited charge carriers too. Therefore, as they stated, “We need to have more in-depth analyses”.

POINT 2: The authors make here a fundamental mistake when describing the effect that a spectral shift due to crystal heating would have. Indeed, for a small spectral shift, we get a TA spectrum that has the form of a derivative where : A) the spectral distance between the PB and PIA peaks relates to the width of the original absorption peak and B) the amplitude of the PB and PIA peaks relates to the amplitude of the spectral shift of the original absorption peak. The case where spectral distance between PB and PIA relates to the spectral shift is for shifts larger than the linewidth: this is not the case here and this would yield TA amplitudes close to unity (not the case here neither). Therefore, considering the data they provide (in Supplementary Figure 7a), we get from A) that the original absorption peak has a width around 20-35 meV, which is the typical width they observe in PL for the exciton lines (Figure2), and from B) that the spectral shift increases almost linearly with

pump fluence, which is expected for crystal heating (with a coefficient around -0.3meV/K). Below is a quick modeling I did to better prove my point. I calculated TA spectra originating from a Lorentzian line (width of 20 meV, dashed curve) with different spectral shifts (noted dE). The

typical data they provide can be quantitatively reproduced. Here the amplitude of the spectral shifts is below -0.05 meV, which corresponds to a rise in temperature below 0.2 K. This lattice temperature rise is even lower than the one the authors claim to reach (up to 30 K). This therefore points out another inconsistency in the overall work that needs to be commented (it can be compared to ref *Nano Lett.* 2017, 17, 2, 644-651, under similar conditions tens of K are reached only at short timescale and only few K or less remain after hundreds of ps).

POINT 3: Here I agree with the authors that many shapes for TA spectra can be obtain when considering the electronic excitation, due to the large sensitivity with the materials conditions of the many-body effects. This corresponds well to the short timescale signals. Yet I point out that the short timescale signals have more variations than the long timescale signals. The latter always present a derivative shape, with a PIA at lower energy than the PB. This is compatible with pure crystal heating (see point 2) and I guess that even better matching could be obtained if broadening of the line due to heating, as well as the evolution (spectral shift and broadening) of other exciton lines (neutral and charged), would be considered.

From this (now quantitative) discussion, a TA signal at long timescale originating simply from pure lattice heating cannot be ruled out and would correspond to the simplest explanation. The main claim about the observation of APR is therefore not sufficiently supported.

Considering that my main concern is still open (and even strengthened with now a quantitative analysis), I maintain my former opinion: I would recommend NOT to publish the manuscript in Nature Communications. Since no better evidence for a “phonon recycling” rather than simple lattice heating has been shown, then the manuscript should be heavily modified to present this narrower claim. It would eventually transform into an interesting, well documented, study on thermal management in 2d heterostructures. I would then be glad to review such revised manuscript.

Response: We are really grateful for the Reviewer 3’s constructive comments. As the Reviewer 3 pointed out, we did make a false estimation on describing the influence of spectral shift on the TA signal. And we completely agree that a very small spectral shift can induce a TA signal with a large energy gap between PIA and PB features. Despite this, there are more experimental data showing that crystal heating is unlikely to be responsible for the long timescale TA signal, as listed below:

- 1、 We are respectively pointing out that Reviewer 3 also made a mistake in the quantitative estimation of the spectral shift. Since the G-WS₂ heterostructure is transferred onto an opaque substrate, namely Si/SiO₂, the reflection curve of the sample is a concave peak at the exciton resonance under a background of unity, rather than a convex peak with a background of zero proposed by the reviewer. According to our calculation, the band shift needs to reach the level of $1\sim 2$ meV to obtain the experimental TA spectrum. Nevertheless, as we mentioned in our previous response, the red shift of the energy band may be caused not only by the temperature rise, but also by the screening of the Coulomb repulsion, when the heat excited the WS₂ carriers. A typical example can be found in TA measurement of perovskite quantum well (DOI: 10.1021/acs.jpcc.5b00148), in which the laser directly excited carriers induce a completely antisymmetric TA spectrum due to many-body interaction rather than lattice heating. Therefore, we emphasize that it is impossible to distinguish the lattice heating and phonon recycling effect simply from the shape of TA spectrum.

2、 Even from a completely symmetric TA signal, the photo-bleaching induced by phonon-excited carriers cannot be safely excluded. Generally, in pump-probe spectroscopy, not only the band shift (ΔE), but also the spectral broadening ($\Delta\sigma$) and bleaching (ΔA), appear under laser excitation. Considering all these factors, the antisymmetric TA signal cannot provide much information. As shown in the figure below, an antisymmetric TA signal naturally appears if we merely consider the band shift, i.e., $\Delta E = -2$ meV, $\Delta\sigma = 0$, and $\Delta A = 0$. However, when including a very strong bleaching effect (caused by phonon recycling effect) of 5% and a slight broadening, i.e., $\Delta E = -2$ meV, $\Delta\sigma = 5$ meV, and $\Delta A = -5\%$, the antisymmetric characteristics of the TA spectrum are well preserved. Therefore, even if there is an antisymmetric TA signal, the carriers induced bleaching effect (i.e., phonon recycling) cannot be ruled out. In other words, an antisymmetric TA spectrum cannot provide much information about the underlying mechanism.

Figure r1: Comparison of the antisymmetric TA spectra induced by different effects

3、 It should be noted that not all the G-WS₂ samples show an antisymmetric TA spectrum at long delay time. The figure below (Supplementary Figure 8) shows the TA spectra from three different samples. The shapes of the spectra are ever-changing whether they are measured at short (~4ps) or long (~400 ps) delay times. The long timescale TA spectrum at the third sample is even completely asymmetric, with a PB peak much stronger than the PA one. This indicates that photo bleaching effects induced by phonon recycling must have a large contribution to the long timescale TA signal, which is completely inconsistent with the crystal heating effect. Therefore, the combination of phonon recycling and many-body effects can explain the TA signals from all G-WS₂ samples, while the pure crystal heating effect cannot. Thus, simply from the signals of the G-WS₂ samples, we can safely confirm the existence of phonon recycling effect. But we are not sure whether lattice heating plays a role.

Supplementary Figure 8: Transition absorption spectra in different samples. (a-c) TA spectra induced by electronic excitation in Sample 1~3. (d-e) TA spectra induced by phononic excitation in Sample 1~3. The PB and PIA features are marked by the blue and red areas. Note that both the amplitude and energy of the PIA feature vary from sample to sample compared to the PB peak.

4. To verify the existence of lattice heating effect, we prepared the graphene-based heterostructure with a different type of TMD (MoS_2), as shown in the figure below (Supplementary Figure 13). Upon excited by 0.4 eV femtosecond pulse, the G- MoS_2 sample shows the same fast response (i.e., electronic response) as that of 2.1 eV direct excitation, which proves that the interlayer coupling of the sample is very good and the effective interlayer charge transfer is realized. Thus, if we follow the idea of lattice heating effect, MoS_2 will inevitably show a slow response (~ 100 ps) caused by band red shift under good interlayer coupling. But the fact is that the G- MoS_2 sample shows a monotonic decay, with no slow TA response expect for the electronic excited one. Furthermore, no antisymmetric TA signal related to the lattice heating induced band shift is found throughout the measured delay time. This indicates that under our low excitation intensity, the influence of lattice heating on the TA spectrum is negligible. On the other hand, the lack of slow response in G- MoS_2 heterostructure can be well explained from the view of phonon recycling. According to the thermal excitation model (Equation 1 in the main text), the thermal excited carrier density is closely related to the Fermi energy level of the TMDs. If the TMD material is nearly intrinsic, the thermal excitation can be ignored, which is the case of G- MoS_2 sample (see discussion in supplementary material Section 8). However, if the sample is doped, where the emission and absorption spectra are dominated by charged trion, the thermal excitation can be sufficiently large for detection (see discussion in the main text). Therefore, considering the TA responses in different graphene-based heterostructures, phonon recycling should prevail over the crystal heating effect, and play a dominant role in the long timescale TA response in G- WS_2 heterostructure.

Supplementary Figure 13: APR effect in an as-prepared G-MoS₂ sample. (a) Optical image of a typical G-MoS₂ sample. (b) 2D pseudo-color TA maps of G-MoS₂ heterostructure under different pump energy and pump regions. Left, controlled MoS₂, $E_{pump} = 2.1$ eV, middle, G-MoS₂, $E_{pump} = 2.1$ eV and right, G-MoS₂, $E_{pump} = 0.48$ eV. (c) Corresponding carrier relaxation dynamics. (d) TA spectra of the G-MoS₂ at different time delays, with pump photon energy of 0.48 eV. (e) Comparison of TA spectra from different sample regions. All the TA spectra are dominated by neutral excitons (A^0 and B^0), indicating an intrinsic doping level of the MoS₂ layer.

5. In fact, the band shift induced by lattice heating effect has negligible impact on the TA response of monolayer TMDs at low excitation power. In the figure below, we pump the monolayer WS₂ with a nearly resonant femtosecond pulse (590 nm, $7 \mu\text{J}/\text{cm}^2$). From this figure we can find that in the whole delay times up to 1 ns, the TA response is always dominated by the PB peak, with a much smaller PIA feature at the lower energy side. Therefore, even if the PIA peak is completely attributed to the lattice heating effect, this effect is negligible compared to the photo-bleaching induced PB feature.

Figure r2: TA response of a monolayer WS₂ excited by a femtosecond pulse with 590 nm, $7 \mu\text{J}/\text{cm}^2$

In summary, based on the whole data we measured, phonon recycling should prevail over crystal heating effect, and play a dominant role in responsible for the long timescale TA response of G-WS₂ heterostructure under sub-bandgap pumping.

Reviewer #3 (Remarks to the Author):

I appreciate the careful consideration of my comments by the authors. The following discussion will account for the 2 response letters they provided (on the website and later through direct communication with the editor), as well as the revised manuscript and supplementary materials.

First, I acknowledge my mistake in the quantitative relation between the lattice temperature rise and the transient optical signal. I do agree that a 1-2 meV redshift can account for the reported experimental signal. What I want to point out is that this shift would correspond to a rise in lattice temperature of around 10K which is the one evaluated by the authors from their thermal relaxation model. In this view, pure lattice heating is enough to explain this one specific transient spectrum (Suppl Fig7).

I do agree with the authors on their claims : "it is impossible to distinguish the lattice heating and phonon recycling effect simply from the shape of TA spectrum" and "an antisymmetric TA spectrum cannot provide much information about the underlying mechanism". But it works both ways: the authors cannot use a specific shape of a TA spectrum to completely rule out lattice heating. In particular, lattice heating not only induces a redshift, but also a broadening (see Nano Lett. 2017, 17, 2, 644-651 for example). Therefore, a spectrally non-antisymmetric transient signal does not rule out lattice heating, in contradiction with Point 3 in authors' response.

Now I focus on the later received letter. The main point here is that the experimental long timescale optical signal is dependent on doping. The authors claim that only phononic excitation can account for such a doping dependence. I disagree. Here I point out that neutral excitons and trions have a slightly different temperature induced redshift (see <https://doi.org/10.1038/s41598-017-14378-w> in MoS₂). More specifically, trions (A⁻ dominant at +50V) have a stronger redshift with temperature, and therefore a stronger transient antisymmetric signal, than neutral exciton (A⁰ appearing at -50V). This is in qualitative agreement with the data reported in Figure2 of the letter. Therefore, lattice heating alone can also explain the data. Here we end up in the similar situation as before where "it is impossible to distinguish the lattice heating and phonon recycling effect simply from the shape of TA spectrum" even with a gate dependence (in contrast to what the authors claim).

Finally, the case with MoS₂ might be the most convincing one and is treated as is in the manuscript (discussion page 7 and Supplementary Section 4 and 8). Yet what is shown is that in this specific case there is no crystal heating visible nor phononic excitation effect (explained here by the neutral doping). It is quite indirect and stretched to extrapolate this one case for all the studied samples. But I acknowledge that this is an argument supporting phononic excitation.

Let me be clear, I follow the authors in their thermal model which leads to a quantitative evaluation of the delayed temperature rise in the TMD lattice after graphene excitation. On itself it is already an interesting study given the many systems and configurations they explored. From this lattice temperature rise in the TMD, I do not doubt that there is a change the carrier density following equation 1 in the manuscript. What I doubt is the authors' claim that this contribution dominates the transient reflection signal. And hence that their results show such a clear evidence for acoustic phonon recycling.

I wish the authors to reflect these elements of doubt in their manuscript.

I would suggest to slightly modify the discussion in page 7 of the manuscript and in Supplementary section 4 following the remarks I made. Also, given the fact that the temperature rise is evaluated from their model, and that the related bandshift and broadening is known from literature, a fare discussion in Suppl section 4 would be to plot the expected transient signal considering only the pure lattice heating. Then the reader could make its own opinion on the relative importance of the other electronic effects. If the authors plan to integrate the data and

discussion in the second response letter (about gate dependence) in their manuscript, I suggest to mention the arguments I presented in my answer above. Also, in lines 247 and 308 of the manuscript, I do not agree with the formulation "confirms the APR effect" and "undoubtedly demonstrates the APR process". In the related discussions, only a good agreement between the calculated temperature rise and the optical signal is demonstrated. Since both lattice heating and phononic excitation signals are proportional to the temperature rise, the authors cannot use these arguments to support APR only. Finally, the discussion around line 343 about the electrical response contribution is quite interesting, yet it is correct only if the optical reflection signal would directly indicate the number of carriers that can be extracted from the system. This is not straightforward (considering lattice heating can participate in the optical signal) and should be clearly mentioned. A study where actual photocurrent is extracted (similar to DOI: 10.1038/ncomms12174) would be the one way to measure this contribution (and to completely rule out my concerns about lattice heating over phononic excitation : I sincerely advice the authors to go in this direction for further works since a positive result would have a major impact in the field).

If the authors were to modify as follow their manuscript, I would support the publication of their work in Nature Communications.

If the authors would not follow the lowering of their claim, then it would be on the editor decision to accept or not this manuscript.

Response Letter

Reviewer #3:

Comment 1:

I appreciate the careful consideration of my comments by the authors. The following discussion will account for the 2 response letters they provided (on the website and later through direct communication with the editor), as well as the revised manuscript and supplementary materials.

First, I acknowledge my mistake in the quantitative relation between the lattice temperature rise and the transient optical signal. I do agree that a 1-2 meV redshift can account for the reported experimental signal. What I want to point out is that this shift would correspond to a rise in lattice temperature of around 10K which is the one evaluated by the authors from their thermal relaxation model. In this view, pure lattice heating is enough to explain this one specific transient spectrum (Suppl Fig7).

I do agree with the authors on their claims : “it is impossible to distinguish the lattice heating and phonon recycling effect simply from the shape of TA spectrum” and “an antisymmetric TA spectrum cannot provide much information about the underlying mechanism”. But it works both ways: the authors cannot use a specific shape of a TA spectrum to completely rule out lattice heating. In particular, lattice heating not only induces a redshift, but also a broadening (see Nano Lett. 2017, 17, 2, 644-651 for example). Therefore, a spectrally non-antisymmetric transient signal does not rule out lattice heating, in contradiction with Point 3 in authors’ response.

Now I focus on the later received letter. The main point here is that the experimental long timescale optical signal is dependent on doping. The authors claim that only phononic excitation can account for such a doping dependence. I disagree. Here I point out that neutral excitons and trions have a slightly different temperature induced redshift (see <https://doi.org/10.1038/s41598-017-14378-w> in MoS₂). More specifically, trions (A- dominant at +50V) have a stronger redshift with temperature, and therefore a stronger transient antisymmetric signal, than neutral exciton (A0 appearing at -50V). This is in qualitative agreement with the data reported in Figure2 of the letter. Therefore, lattice heating alone can also explain the data. Here we end up in the similar situation as before where “it is impossible to distinguish the lattice heating and phonon recycling effect simply from the shape of TA spectrum” even with a gate dependence (in contrast to what the authors claim).

Finally, the case with MoS₂ might be the most convincing one and is treated as is in the manuscript (discussion page 7 and Supplementary Section 4 and 8). Yet what is shown is that in this specific case there is no crystal heating visible nor phononic excitation effect (explained here by the neutral doping). It is quite indirect and stretched to extrapolate this one case for all the studied samples. But I acknowledge that this is an argument supporting phononic excitation.

Let me be clear, I follow the authors in their thermal model which leads to a quantitative evaluation of the delayed temperature rise in the TMD lattice after graphene excitation. On itself it is already an interesting study given the many systems and configurations they explored. From this lattice temperature rise in the TMD, I do not doubt that there is a change the carrier density following equation 1 in the manuscript. What I doubt is the authors’ claim that this contribution dominates the transient reflection signal. And hence that their results show such a clear evidence for acoustic phonon recycling.

I wish the authors to reflect these elements of doubt in their manuscript.

Response 1:

We really appreciate Reviewer 3's constructive comments. We completely agree that it is very hard to distinguish the crystal heating and phononic excitation simply from the optical response of the heterostructure. Maybe an actual G-WS₂ device for photocurrent (PC) extraction is the only way to completely separate them, as the crystal heating cannot provide actual photo carriers. However, the PC measurement would also introduce new problems. Specifically, the steady state PC measurement cannot resolve the contribution from the ultrafast electronic excitation (i.e., carrier transfer) and the slow phononic excitation, so a time-resolved measurement is required, and that is what we will go ahead with.

At the current stage, the crystal heating and phononic excitation are indeed impossible to distinguish, so in the new manuscript, the crystal heating effect is no longer completely ruled out (See page 7 in the manuscript and Supplementary section 4). Instead, the question is still open to the readers. Nevertheless, we still claim that there are two indirect experimental data showing that the phononic excitations should have a contribution to the slow rising TA signal: the various TA spectral shapes of different G-WS₂ samples, and the absence of the slow TA feature in G-MoS₂ sample. At the same time, we no longer exclude the existence of crystal heating effect in G-WS₂ samples.

Comment 2:

I would suggest to slightly modify the discussion in page 7 of the manuscript and in Supplementary section 4 following the remarks I made. Also, given the fact that the temperature rise is evaluated from their model, and that the related bandshift and broadening is known from literature, a fare discussion in Suppl section 4 would be to plot the expected transient signal considering only the pure lattice heating. Then the reader could make its own opinion on the relative importance of the other electronic effects. If the authors plan to integrate the data and discussion in the second response letter (about gate dependence) in their manuscript, I suggest to mention the arguments I presented in my answer above.

Also, in lines 247 and 308 of the manuscript, I do not agree with the formulation “confirms the APR effect” and “undoubtedly demonstrates the APR process”. In the related discussions, only a good agreement between the calculated temperature rise and the optical signal is demonstrated. Since both lattice heating and phononic excitation signals are proportional to the temperature rise, the authors cannot use these arguments to support APR only.

Finally, the discussion around line 343 about the electrical response contribution is quite interesting, yet it is correct only if the optical reflection signal would directly indicate the number of carriers that can be extracted from the system. This is not straightforward (considering lattice heating can participate in the optical signal) and should be clearly mentioned. A study where actual photocurrent is extracted (similar to DOI: 10.1038/ncomms12174) would be the one way to measure this contribution (and to completely rule out my concerns about lattice heating over phononic excitation : I sincerely advice the authors to go in this direction for further works since a positive result would have a major impact in the field).

If the authors were to modify as follow their manuscript, I would support the publication of their work in Nature Communications.

If the authors would not follow the lowering of their claim, then it would be on the editor decision to accept or not this manuscript.

Response 2:

We thank Reviewer 3 for his/her evaluation and insightful comments. We have made several changes in the new manuscript. Firstly, page 7 of the main text and Supplementary section 4 have been revised according to reviewer 3. In brief, we claim that the possibility of crystal heating cannot be proved or excluded simply from the optical response. But we propose that phononic excitation should work in the slowing TA response of G-WS₂ heterostructure. The debate regarding crystal heating and phononic excitation is still open to readers. Moreover, in the detailed discussion in Supplementary section 4, the quantitative estimation of both the band shift induced by crystal heating and the photo bleaching caused by phononic excitation has been carried out, followed by their comparison with the experimental data. Just as Review 3 suggested, the readers can make their own decision on these two mechanisms. In addition, the gate dependence TA response is not intended to be included in the new supplementary data, as the gate efficiency is still very poor. An improvement of the device fabrication process should be made to enhance the gate efficiency.

Secondly, the expression of “confirms the APR effect” in line 247 has been modified to “confirms the thermal transport from graphene to WS₂”, and the statement of “undoubtedly demonstrates the APR process” has been changed to “undoubtedly demonstrates the interlayer thermal transport.”

Finally, the discussion at around line 343 has been modified: “If the crystal heating effect is negligible, the slow rising TA feature can be completely attributed to the APR process, which will play a more important role (i.e., ~50 times) than the ultrafast carrier transfer when considering the temporal integration of the reflectance signal. Therefore, further study on the actual photocurrent response under sub-bandgap photon excitation is required, which is not only of theoretical significance, but also useful for the optimization of practical device efficiency.”

REVIEWERS' COMMENTS:

Reviewer #3 (Remarks to the Author):

I thank the authors for the constructive discussion throughout the review process. In view of the last modifications made to the manuscript and supplementary material, I support the publishing of this work in Nature Communications.

Response Letter

Reviewer #3:

Comment 1:

I thank the authors for the constructive discussion throughout the review process. In view of the last modifications made to the manuscript and supplementary material, I support the publishing of this work in Nature Communications.

Response 1:

We really appreciate Reviewer 3's constructive comments and suggestions that help improve the quality of this manuscript. We have tried our best to modify our work, and we are very happy that the hard efforts pay off. Finally, we want to thank Reviewer 3 again for his/her support of publishing our work in Nature Communications.